# Observed Water- and Light-Limitation Across Global Ecosystems

François Jonard[1,2], Andrew F. Feldman[3,4], Dan J. Short Gianotti[5], Dara Entekhabi[5]

[1] Earth Observation and Ecosystem Modelling Laboratory, SPHERES Research Unit, Université de Liège (ULiege), 4000 Liège, Belgium
[2] Agrosphere (IBG-3), Institute of Bio- and Geosciences, Jülich Research Centre, Jülich, Germany
[3] NASA Postdoctoral Program, NASA Goddard Space Flight Center, Greenbelt, MD, USA
[4] Biospheric Sciences Laboratory, NASA Goddard Space Flight Center, Greenbelt, MD, USA
[5] Parsons Laboratory, Department of Civil and Environmental Engineering, Massachusetts Institute of Technology (MIT), Cambridge, MA, USA

*Correspondence to*: François Jonard (francois.jonard@uliege.be)

**Abstract.** With a changing climate, it is becoming increasingly critical to understand vegetation responses to limiting environmental factors. Here, we investigate the spatial and temporal patterns of light and water limitation on photosynthesis using an observational framework. Our study is unique in characterizing the nonlinear relationships between photosynthesis and water and light, acknowledging approximately two regime behaviors (no limitation and varying degrees of limitation). It is also unique in using an observational framework instead of using model-derived photosynthesis properties. We combine data from three different satellite sensors, i.e., sun-induced chlorophyll fluorescence (SIF) from TROPOMI, surface soil moisture from SMAP, and vegetation greenness from MODIS. We find both single-regime and two-regime models describe SIF sensitivity to soil moisture and photosynthetically active radiation (PAR) across the globe. The distribution and strength of soil moisture limitation on SIF are mapped in the water-limited environments while the distribution and strength of PAR limitations are mapped in the energy-limited environments. Two-regime behavior is detected in 73% of the cases for water limitation on photosynthesis, while two-regime detection is much lower at 41% for light limitation on photosynthesis. SIF sensitivity to PAR strongly increases along moisture gradients, reflecting mesic vegetation's adaptation to making rapid usage of incoming light availability on the weekly timescales. The transition point detected between the two regimes is connected to soil type and mean annual precipitation for the SIF-soil moisture relationship and for the SIF-PAR relationship. These thresholds have therefore an explicit relation to properties of the landscape, although they may also be related to finer details of the vegetation and soil interactions not resolved by the spatial scales here. The simple functions and thresholds are emergent behaviors capturing the interaction of many processes. The observational thresholds and strength of coupling

can be used as benchmark information for Earth system models, especially those that characterize gross primary

production mechanisms and vegetation dynamics.

## 1 Introduction

Vegetation plays a large role in the Earth system, modulating land-atmosphere exchanges of water, carbon, and energy (Beer et al., 2010; Jasechko et al., 2013). With increasing temperatures and changing precipitations, and possibly more intense drought and heatwaves, these change-induced factors that affect vegetation productivity have impacts on global carbon budgets and food security (Liang et al., 2017; Huang et al., 2018; Gentine et al., 2019). It is therefore imperative to understand how vegetation function responds to environmental factors across the globe. More specifically, it is important to understand how climatic factors create limitations on vegetation function at large spatial scales (Ahlström et al., 2015; Zhang et al., 2020a,b; Li et al., 2021). Such determinations using observed datasets are key for predicting and validating terrestrial ecosystem productivity responses in Earth system models (Fisher et al., 2018), ultimately improving our ability to predict the future land surface conditions in the context of global change.

Remote sensing has proven to be a useful tool for mapping and monitoring vegetation function across the globe. Satellite observations provide the ability to spatially integrate over the behavior of whole ecosystems, providing scaled-up behavior relevant to the global carbon cycle and Earth system models. Observations of sun-induced chlorophyll fluorescence (or commonly called solar-induced fluorescence; SIF)—radiation emitted at wavelengths of 650 to 800 nm from plant photosystems—are valuable indicators of ecosystem photosynthetic activity. In contrast to traditional vegetation reflectance indices, SIF is sensitive to diurnal and seasonal photosynthetic dynamics and not only to changes in greenness (Wang et al., 2020). SIF emission is connected to transpiration and photosynthesis-related processes and these relationships are controlled by intrinsic water use efficiency (WUE) and light use efficiency (LUE). Recent studies have shown the value of satellite observations of SIF to monitor ecosystem transpiration (Lu et al., 2018; Pagan et al., 2019; Shan et al., 2019; Maes et al., 2020) and productivity (gross primary production, GPP) (Joiner et al., 2014; Zhang et al., 2016; He et al., 2020). Since 2009, surface soil moisture (SM) can also be derived globally from low-frequency microwave (L-band; 1.4 GHz) radiometer observations (Entekhabi et al., 2010; Kerr et al., 2010). While microwave measurements are sensitive to the water in the top 5 centimeters of the soil profile, it has been shown that SM estimates averaged over several days are both physically and statistically correlated to deeper root zone soil moisture (Akbar et al., 2018b; Short Gianotti et al., 2019b; Feldman et al., 2022).

Using these satellite remote sensing developments, several studies have analyzed the influence of bio-climatic factors on productivity. Madani et al. (2017) found that reanalysis-derived soil moisture (SM), vapor pressure deficit (VPD), and minimum daily air temperature are significant control factors influencing ecosystem productivity over the globe. They showed that SIF was positively correlated with soil moisture on monthly time-scales in dry biomes (e.g., Sahel), whereas in humid biomes (e.g., Amazonia), SIF was negatively correlated with soil moisture and positively correlated to VPD. While Global Ozone Monitoring Experiment-2 (GOME-2) satellite SIF observations were used as proxy of productivity, environmental factors were derived from model reanalysis data, which may have model-prescribed relationships between one another and with productivity. In addition, factors influencing vegetation growth were limited to temperature and moisture constraints, but other environmental controls such as light limitation were not addressed.

Similarly, Nemani et al. (2003) have analyzed the impact of global climate changes on vegetation productivity using global reanalysis data and a production efficiency model. Their results indicate differential controls of light, water, and temperature on vegetation, but with mainly a reduction of climatic constraints to plant growth during the last two decades of the past century, with significant increase in net primary production over large regions of Earth such as in Amazon rain forests. Walker et al. (2020) review theory and evidence suggesting a substantial increase in global photosynthesis since pre-industrial times driven at least in part by increased atmospheric carbon dioxide concentration leading to increases in plant water use efficiency.

Recently, using Orbiting Carbon Observatory-2 (OCO-2) SIF and Soil Moisture Active and Passive (SMAP) SM satellite observations, Gonsamo et al. (2019) found that SM was often a primary limiting factor to plant growth in drylands and croplands. While based on a low number of concurrent SIF and SM data records, the authors observed positive and stronger SIF-SM relationships in drier and warmer regions. In their study, nonlinear behaviors were not addressed.

Using satellite observations of SIF and climate datasets, Liu et al. (2020) found that SM has a dominant role in determining dryness stress on ecosystem production over most land vegetated areas. However, the study was primarily interested in moisture effects, having investigated the relative role of SM and VPD in limiting ecosystem production. Short Gianotti et al. (2019a) showed that SIF-SM relationships match satellite-derived GPP-SM relationships in both time and space, with little-to-no SIF-SM relationship in the light-limited humid regions of the contiguous United States and increasing response strength with aridity. Water-limited regions showed strong increases in ecosystem-water use efficiency (daily SIF or GPP divided by latent heat flux) during SM dry spells.

Studies investigating global drivers of photosynthesis tend to focus on linear relationships between these variables, which potentially neglects nonlinear conditions where photosynthesis is not-limited (Teubner et al., 2018; Gonsamo et al., 2019). For example, the light use efficiency (LUE) model is widely used in Earth system modeling to simulate GPP as a linear function of absorbed light (Monteith, 1972). It is becoming more evident that nonlinear plant function behavior exists, especially depending on soil moisture (under dry and wet moisture states) (Feldman et al., 2018; Short

Gianotti et al., 2019a; Bassiouni et al., 2020). Those that evaluate nonlinear relationships do so regionally or globally and do not evaluate on a per-pixel basis (Madani et al., 2017). The global patterns of SIF relationships with water and light across climates and biomes remain under-characterized. Yet, the influence of environmental factors on vegetation productivity (and carbon cycle) has both weather timescale and seasonal timescale (relative timing of warm and wet seasons). Both timescales are important and exist in nature. Ecosystem responses on these different timescales are to

date not well understood (Linscheid et al., 2020).

The objective of this study is to evaluate the environmental factors that limit surface water and carbon exchanges over vegetated areas. Specifically, we ask: what are the conditions under which SIF is limited by water and light in space and time? Can we detect first-order nonlinear controls of water and light on photosynthesis as suggested from theory? If so, are there climatic controls on the water and light availability thresholds that divide regimes of SIF nonlinear

responses to environmental variables? Here, we use an observation-based framework to evaluate nonlinear relationships between SIF and available water and light. Observations are key to provide benchmark information for parameterizing effects of water-stress or light-limitation on ecosystem productivity in Earth system models. Modeled vegetation products can implicitly or explicitly parameterize the relationships between SIF and water and light that we intend to evaluate. In this observationally driven study, we combine three data streams — sun-induced chlorophyll

fluorescence (SIF) from TROPOMI, surface soil moisture from SMAP, and normalized difference vegetation index (NDVI) from MODIS—to globally monitor observational evidence for seasonal water-limitation and light-limitation in plant function.

## 2 Data

Satellite-based data were collected and analyzed for our main per-pixel approach for a 2.5-year period from April 2018

to September 2020 (determined by the concurrently available TROPOMI and SMAP data; see Table 1). Climatology information from decade-long time series were used as auxiliary datasets.

### 2.1 Global Satellite Data

### 2.1.1 TROPOMI Sun-Induced Chlorophyll Fluorescence

Sun-induced chlorophyll fluorescence (SIF, mW m$^{-2}$ nm$^{-1}$ sr$^{-1}$) data are obtained from the TROPOspheric Monitoring Instrument (TROPOMI) aboard the Sentinel-5 Precursor satellite (Köhler et al., 2018). TROPOMI provides optical observations with a spectral resolution of 0.5 nm, a spatial resolution of 7 x 3.5 km² (along track x across track) at nadir, and almost global coverage within 1 day. Sentinel-5 Precursor has an overpass time near 13:30 local solar time. SIF is retrieved in a spectral window ranging from 743 to 758 nm using the method of Köhler et al. (2018). SIF observations with large cloud cover (cloud fraction larger than 0.8) were filtered out as described in Köhler et al. (2015). As a robustness test, we additionally use SIF data from the Global Ozone Monitoring Experiment-2 (GOME-2) instrument aboard the MetOp-A satellite, on the period of April 2015 and March 2019, the longest period for which SMAP and GOME-2 are jointly available (Joiner et al., 2013).

### 2.1.2 SMAP Soil Moisture

Surface soil moisture (SM, m³ m$^{-3}$) data (top 5 cm) are from the L-band (1.4 GHz) microwave radiometer aboard the NASA Soil Moisture Active/Passive (SMAP) satellite (Entekhabi et al., 2010). Microwave observations from the 6 a.m. descending overpasses were used with a spatial resolution of 36 x 36 km² and a global coverage within 3 days. Retrievals of soil moisture were obtained using the multi-temporal dual channel algorithm (MT-DCA) (Konings et al., 2016; Feldman et al., 2021). The MT-DCA algorithm estimates vegetation attenuation and scattering from an algorithm with temporal regularization. It does not use any information on land-use and ecosystem classifications which would bias the results otherwise. While the microwave measurements are commonly known to reflect the top 5 cm, several lines of evidence suggest SM can viably represent relevant root zone dynamics in most cases. First, under wetter conditions, SMAP SM is known to closely correlate with rootzone dynamics, especially in the upper 50 cm (Akbar et al., 2018b; Short Gianotti et al., 2019b). Under drier conditions, microwave emission depth originates from deeper than 5 cm, down to a meter in some cases depending on soil properties (Njoku and Entekhabi, 1996). Furthermore, many plants, especially species in semi-arid grasslands where we mainly evaluate SIF-SM, have rooting distributions skewed to the upper layers (<30cm) with preferential uptake of water in the upper soil layers (Flanagan et al., 1992; Meinzer et al., 1999; Miguez-Macho and Fan, 2021). As such, SMAP, in fact, effectively senses soil layers deeper than 15-25 cm, relevant to global root water uptake especially in water-limited ecosystems (Feldman et al., 2022).

### 2.1.3 MODIS Normalized Difference Vegetation Index (NDVI)

Normalized Difference Vegetation Index (NDVI) data come from the Moderate Resolution Imaging Spectroradiometer (MODIS) instrument aboard the NASA Terra satellite. NDVI data between January 1, 2003 and December 31, 2021 were obtained from the Level 3 MODIS product which is a cloud-free 16-day 0.05 degree dataset (MOD13C1, https://lpdaac.usgs.gov/products/mod13c1v006/). We linearly resampled the 0.05° grid to the 36 km EASE2 grid. Then, we linearly interpolated the 16-day NDVI data to daily values to determine growing season start and end dates within each pixel.

### 2.2 Ancillary Data for Analyses

### 2.2.1 Mean Annual Precipitation and Soil Type

Mean annual precipitation is obtained by averaging annual means between 2010 and 2020 from the Integrated Multi-satellitE Retrievals for GPM (IMERG) final run product combining data from the Global Precipitation Measurement (GPM) satellite constellation (Huffman et al., 2019). Sand and clay fraction information was also obtained from the SoilGrids250m database (Hengl et al., 2017). These metrics were used to evaluate climate gradients of spatial maps generated in the analysis.

### 2.2.2 MERRA-2 Downwelling Photosynthetically Active Radiation

Daily and downwelling photosynthetically active radiation (PAR, W m$^{-2}$) data are provided by the NASA Modern-Era Retrospective analysis for Research and Applications, Version 2 (MERRA-2) global reanalysis (GMAO, 2015). The spatial resolution is 0.5° x 0.625°. While PAR is an observation-driven modelled product, it is not expected to have strong relationships with vegetation function prescribed within the model because it is driven mainly by solar seasonality and assimilated atmospheric fields such as cloud cover.

### 2.3 Spatial and Temporal Aggregations

The SIF, PAR and NDVI data were regridded on a linear weighting basis to the EASE-2 SMAP grid (36 x 36 km²). The SIF, SM and PAR data were also aggregated temporally to produce 8-day composites. The temporal aggregation was performed to smooth SIF data, with 8-day aggregation selected to match the exact repeat cycle of SMAP. To increase sample size for the correlation maps and the regime classification, the SIF, SM and PAR data at 36 x 36 km² resolution are pooled in 2 x 2 pixel boxes, for each 8-day period. In the following section, all maps have therefore a spatial resolution of 72 x 72 km².

**Table 1. List of datasets used in this study with their respective native spatial and temporal resolution. Note that the datasets were all linearly aggregated to a spatial resolution of 72 x 72 km² and 8-day periods.**

| Variable | Source of data | Spatial resolution | Temporal resolution |
|---|---|---|---|
| Sun-induced chlorophyll fluorescence | Sentinel-5P satellite TROPOMI instrument | 7 x 3.5 km² | Daily |
| Soil moisture | SMAP satellite L-band radiometer instrument | 36 x 36 km² | 3 days |
| Normalized Difference Vegetation Index | Terra satellite MODIS instrument | 0.05° | 16 days |
| Photosynthetically active radiation | MERRA-2 global reanalysis | 0.5° x 0.625° | Daily |
| Precipitation | GPM satellite constellation IMERG product | 0.1° x 0.1° | Half-hourly |

## 3 Methodology

### 3.1 Growing Season Estimation

Since we analyze the seasonal water-limitation and light-limitation of plant function only during the growing season, the growing season for each global pixel was first defined using NDVI climatology. The NDVI climatology was developed by averaging 19 years (2003 to 2021) of NDVI data into a mean climatology and smoothing using a 90-day moving average filter (Fig. S1). The growing season was defined by first finding the peak of the NDVI climatology to identify the main growing season and then finding the green up and brown down times as when NDVI reaches its median before and after this peak. This results in growing seasons with a peak on DOY between 100 to 275 in the northern hemisphere and DOY typically 0 to 50 and 300 to 365 in many regions of the southern hemisphere (Fig. 1). There are a number of different approaches to estimating plant phenology based on satellite measurements (e.g., see Moulin et al., 1997; Zhang et al., 2006; Bush et al., 2018; Morellato et al., 2018; Peano et al., 2019). Ultimately the applied technique depends on the application needs, and the approach followed here is sufficient to characterize the active growing season encompassing the primary water and energy interactions with the carbon cycle. It is worth noting that for some regions, several peaks could be observed. The peak with the maximum NDVI was selected corresponding to the "primary growing season", such as in the tropics, which are characterized by a smaller seasonal amplitude. We additionally tested shorter growing seasons to ensure that shoulder seasons did not influence results.

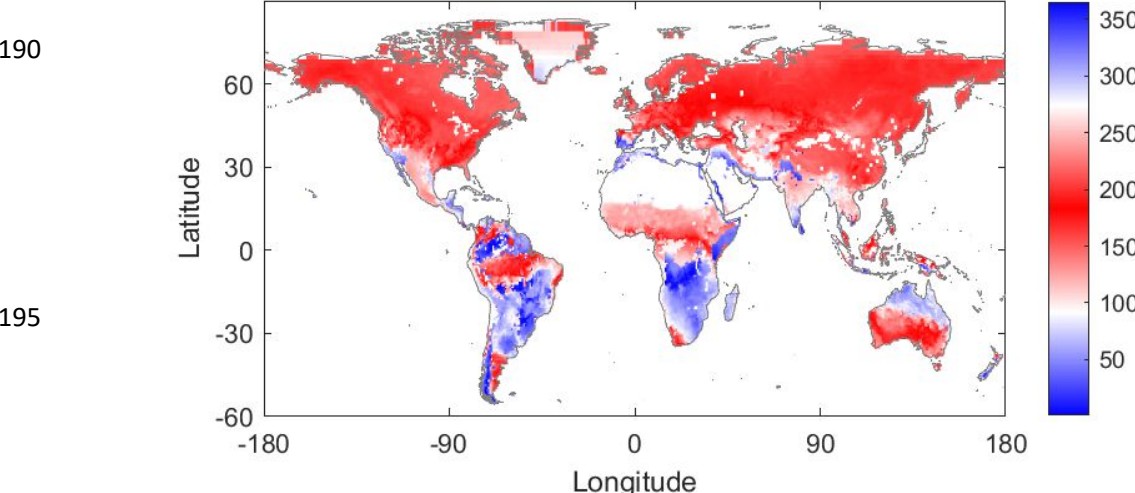

**Figure 1: Growing season determination based on MODIS satellite normalized difference vegetation index (NDVI). Day of the year for the phenological peak based on MODIS NDVI climatology. White shading indicates NDVI was not available (bare-soil or water bodies).**

### 3.2 Conceptual Basis

While chlorophyll fluorescence originates from energy partitioning at the photosystem (leaf) level, SIF presents as an aggregated landscape variable at large spatial and temporal scales (as considered in this study). It is a function of the absorbed PAR ($APAR_{Chl}$ = PAR x $fPAR_{Chl}$, with $fPAR_{Chl}$ being the fraction of PAR absorbed by chlorophyll pigments and mainly controlled by vegetation cover fraction, leaf area index, leaf chlorophyll content, and plant structure) and the fluorescence quantum yield ($\phi_F$, mainly controlled by leaf biochemical properties and is dependent on plant health and water status), both correlated with photosynthetic productivity and strongly influenced by water and energy (light) availability (Joiner et al., 2014; Jonard et al., 2020; Magney et al., 2020):

$$SIF(\lambda) = PAR \times fPAR_{Chl} \times \phi_F(\lambda) \times f_{esc}(\lambda) \times \tau_{atm}(\lambda) , \tag{1}$$

with $f_{esc}$ being the fraction of SIF (at wavelength $\lambda$) emitted from all leaves that escape from the canopy and $\tau_{atm}$ being the fraction of SIF that also passes through the atmosphere ($\tau_{atm}$).

The behavior of the factors in Eq. (1) differs strongly throughout the globe. For instance, annual croplands tend to show large variations in $fPAR_{Chl}$ and $f_{esc}$ during the growing season, while these factors are expected to remain more constant over evergreen forests. The value of $\phi_F$ is expected to react to the ambient stress conditions (De Cannière et al., 2021). Water- or light limitation comprises the combination of all these components. Light- and water limited photosynthesis will first impact the photosynthetic machinery, affecting $\phi_F$. A prolonged water or light limited regime

will manifest in primary production and biomass growth, and therefore on fPAR$_{Chl}$ and on f$_{esc}$. Evaluating the combination of the parameters in Eq. (1) provides insights on the limiting factors of the plant growth. Furthermore, it

is expected that many of these parameters are nonlinearly related to water and light limitation (Xu et al., 2021).

We emphasize that it is not our goal to investigate all possible limiting factors on photosynthesis (e.g., temperature, nutrient limitation), nor how they interact to create states where one or both variables are limiting. A more comprehensive analysis can classify states along multiple axes of climatic factors. Our single-axis classification provides a first step towards such classifications in detecting globally where, to a first-order, nonlinear relationships

between SIF and water and/or light emerge. It also determines observed spatial variations of the types of climatic factor relationships with SIF. The effects of water and light are at least expected to capture the major global limiting pathways based on previous work (Madani et al., 2017).

### 3.3 Correlation Maps

As a zeroth-order analysis motivating our subsequent evaluation of SIF, we first evaluate the Pearson correlation

coefficient between SIF and SM (Fig. 2) and SIF and PAR (Fig. 3) using our 8-day aggregations of each variable. Only values from the growing season are used. In this case, factors that directly limit SIF appear as positively correlated with SIF (in blue in Fig. 2 and 3). Ultimately, the correlation maps guide subsequent analysis of more detailed two-regime behavior. The SIF-SM correlation map (Fig. 2) shows large regions of water limitation (blue regions), such as the Sahel, Eastern and Southern Africa, Eastern Brazil, Southern Asia, and Eastern Australia. The SIF-PAR correlation

map (Fig. 3) shows large regions of positive correlation (blue regions), such as much of the United States, Southern Brazil, Europe, and Russia, which are negatively correlated with SM in Fig. 2. Green et al. (2017) provide a more in-depth analysis of linear correspondences between vegetation growth and land/atmosphere variables among others.

Low SIF-SM correlations occur mainly in densely forested regions where soil moisture estimation uncertainty is largest and potentially surface soil moisture is less of a control on vegetation function than other factors. Regions in blue in

Fig. 2 are generally in red in Fig. 3, revealing that SM and PAR are typically negatively correlated (Fig. S2). This is due to synoptic-scale correlations between cloud cover and soil moisture (positive) and between cloud cover and shortwave radiation (negative), as well as seasonal-scale alignment of growing-season peaks with peaks in the primarily-limiting component: light or water. Given these considerations, in the remainder of our analysis, we do not evaluate negative relationships between SM (or PAR, separately) and SIF, which could reflect a spurious relationship

due to limitation from the other variable. We expect that positive correlations between SM (or PAR) and SIF reflect

causal relationships from SM (or PAR) to photosynthesis in limiting environments. We similarly hypothesize that negative correlations between SM and SIF indicate light-limitation or a lagged response of root uptake and increased PAR following precipitation events.


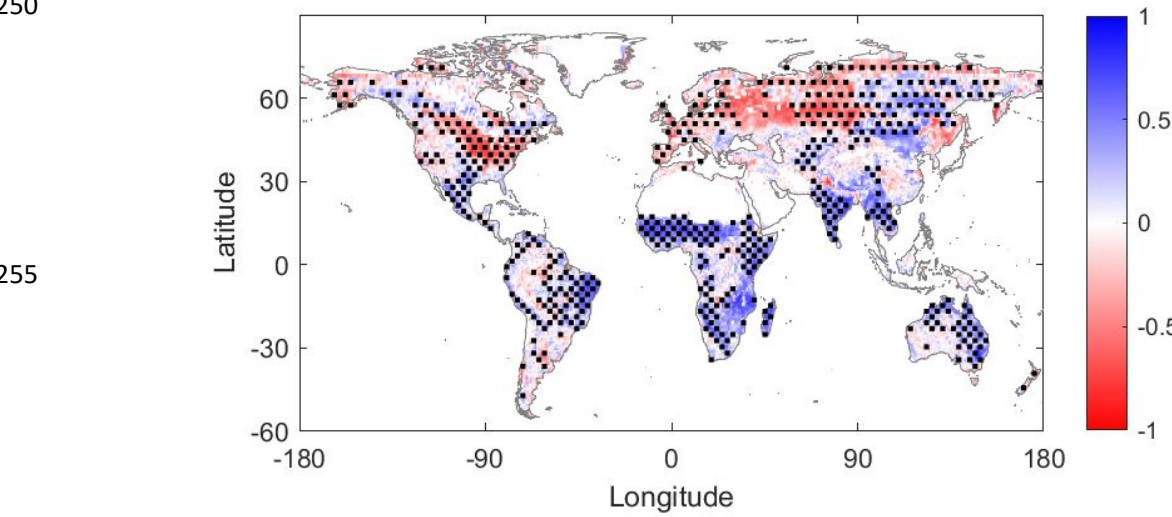



**Figure 2: TROPOMI sun-induced chlorophyll fluorescence (SIF) and SMAP MT DCA soil moisture (SM) growing season correlation. Pearson correlation coefficient of 8-day averages. Regions of statistical significance (*P* < 0.05) are indicated with stippling. The stippling corresponds to distributed areas of statistically significant grid points and therefore each statistically significant pixel may not have its own stipple symbol.**


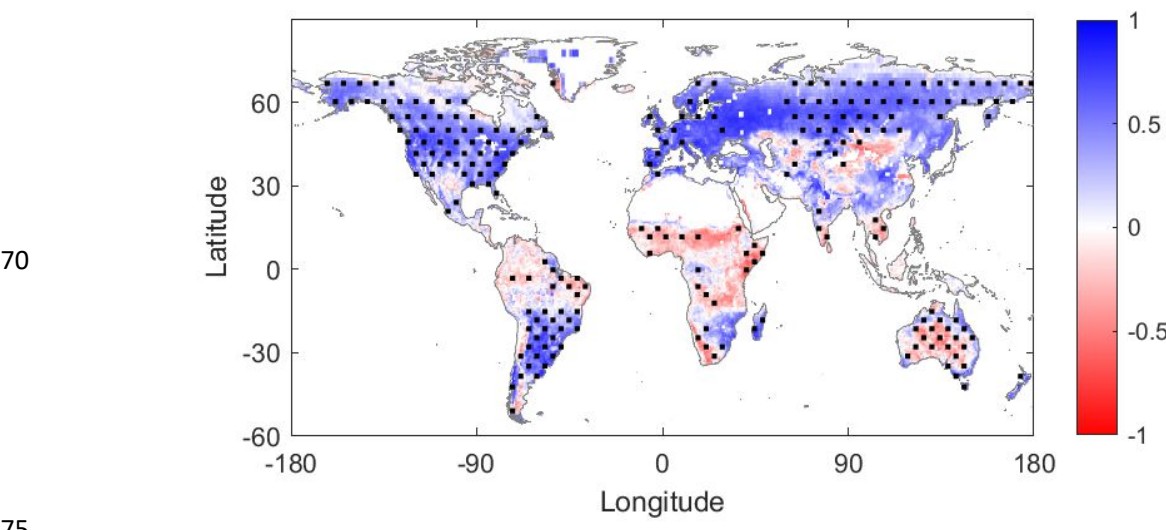



**Figure 3: TROPOMI sun-induced chlorophyll fluorescence (SIF) and MERRA-2 photosynthetically active radiation (PAR) growing season correlation. Same conventions as Fig. 2.**

### 3.4 Regime classification

Pearson correlation provides information about the degree to which a variable linearly limits SIF. However, in many cases, nonlinear relationships are present where the strength of limitation may decrease above a certain threshold of soil moisture or photosynthetically active radiation (e.g., see Fig. 5). This therefore can bias linear correlations and obscure their interpretation in Figs. 2 and 3 as well as previous studies (Gonsamo et al., 2019). We approximate this relationship here as a two-regime linear model to characterize conditions when water or light limits SIF. In some

instances, only one regime may be observed for either water or light. Therefore, three distinct models were tested representing three scenarios for each limitation (water and light) (Fig. 4) as in previous studies that evaluated surface energy fluxes (Akbar et al., 2018a; Feldman et al., 2019).

The following models are used for SM and PAR separately and independently. Only model selections are completed in the pixels where the given variable is positively correlated with SIF (Figs. 2 and 3). If a given pixel shows a positive

SIF correlation with both SM and PAR, then models for both SM and PAR are estimated. The first model is the linear model representing the water- or light-limited regime (Fig. 4a and d). Here, the conditions are always characterized by water- or light-limitation without another regime of behavior detected. An increment of SM or PAR always impacts photosynthesis and therefore the SIF. The second model is the full two-regime model representing the two-regimes of water- and light-limitation (Fig. 4b and e). Only when this regime is determined a moisture or light threshold will be

estimated. Below this threshold, the given variable limits SIF. Above the threshold, an increment of SM or PAR will not affect photosynthesis. The third model is the zero-slope model for the no water or no light limitation regime (Fig. 4c and f). In this case, plant growth is not sensitive to water or light within the variability observed at that location.

A model is selected based on the Bayesian Information Criterion (BIC) in order to avoid over-fitting among models shown in Fig. 4. Specifically, BIC penalizes more complex models (i.e., two regime model here) that inherently

increase the model fit to the data, but may not provide more predictability than more parsimonious models. For example, the linear model has one additional parameter than the zero-slope model, which is the slope. In order for the linear model to be selected, the additional parameter of a non-zero slope must increase the model fit beyond the penalty of overfitting. Note that information criterion like BIC have been applied to similar model selection applications of water and energy limitation (Feldman et al., 2019; Schwingshackl et al., 2017). Example pixels are shown in Fig. 5

where the two-regime model is selected by this method. Note that we only perform the analysis on pixels with at least 20 pairs of SIF-SM or SIF-PAR.

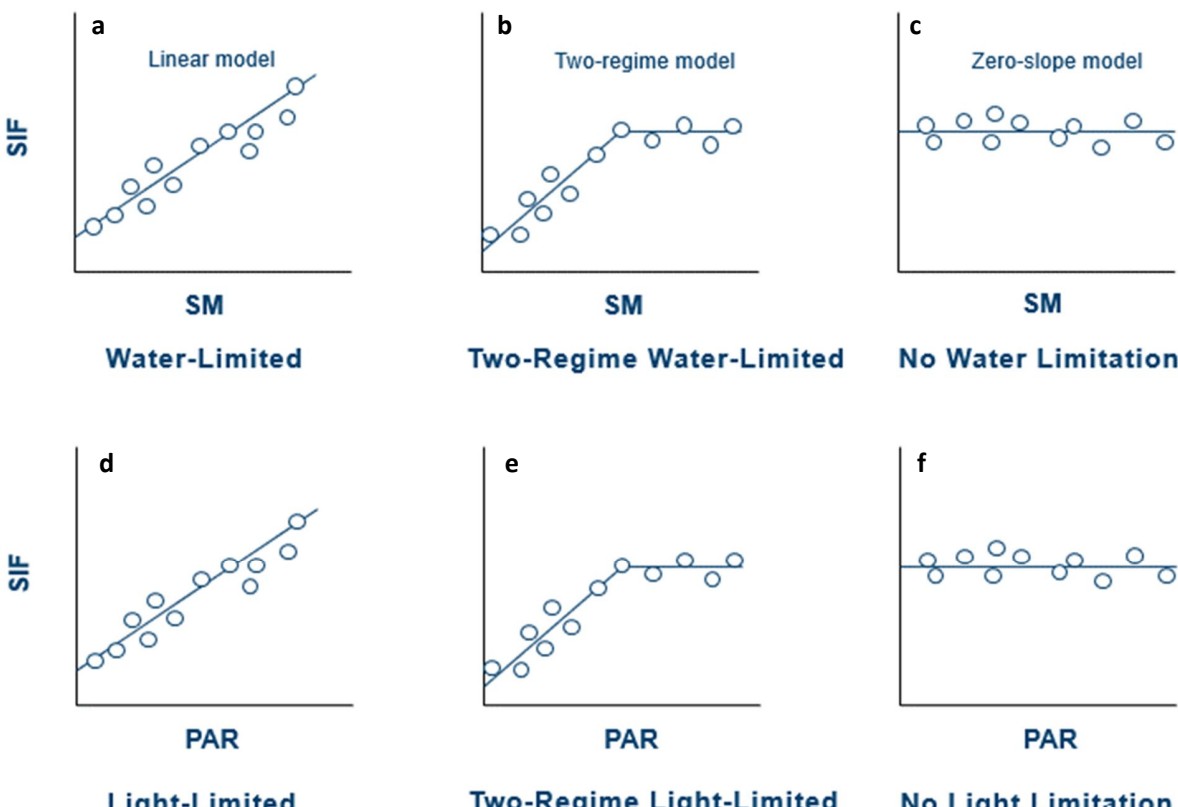

**Figure 4: Schematics of model types for the SIF-SM and SIF-PAR regimes. Bayesian Information Criterion (BIC) is used to avoid over-fitting among models during statistical selection.**

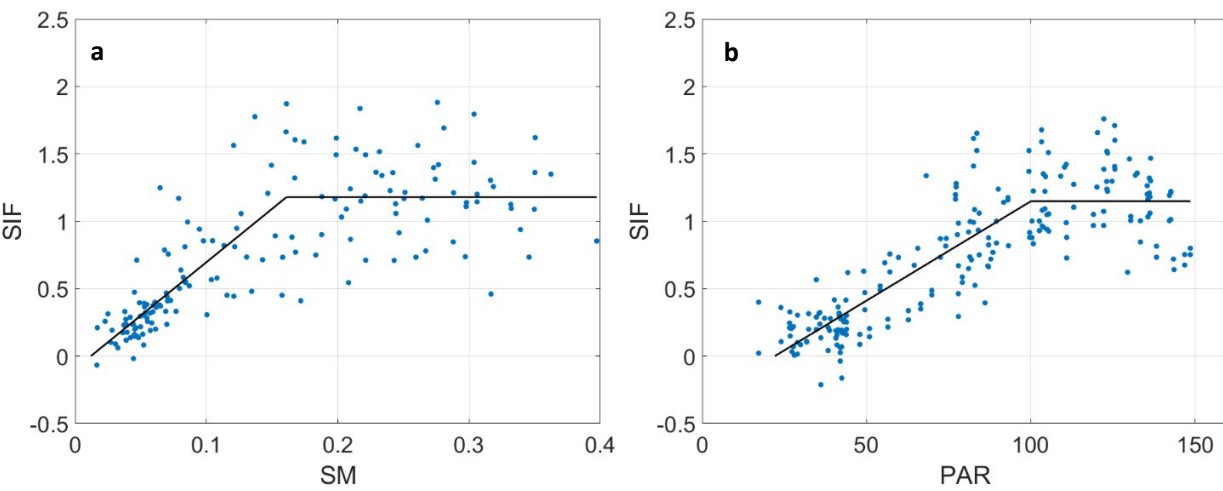

**Figure 5: Example SIF relationships with water and light with model estimation from Fig. 4 demonstrated. Scatter plot of (a) sun-induced chlorophyll fluorescence (SIF, [mW m$^{-2}$ nm$^{-1}$ sr$^{-1}$]) and soil moisture (SM, [m$^3$ m$^{-3}$]) data, and (b) SIF and photosynthetically active radiation (PAR, [W m$^{-2}$]) data for the primary growing season and for a single 72 x 72 km² pixel**


located in (a) Sahel region (Mali, latitude/longitude: 13.68°N/6.35°W) and (b) Spain (Zamora, latitude/longitude: 41.25°N/5.60°W). Data are fitted with a two-regime model.

## 4 Results and Discussion

In this section, we present the results of the selection of the best-fit model (based on the Bayesian information criterion) among the three model types described above (linear, two-regime or zero-slope model) for each factor's (water and light) limitation on SIF.

### 4.1 Water-limited regimes

The spatial distribution of the selected model types, the corresponding model slopes, and the frequency distribution of model threshold for the SIF-SM relationship are shown in Fig. 6. Several regions with a two-regime water-limitation can be clearly identified, such as most of sub-Saharan Africa (except the Congo Basin), Southern Asia, Eastern Australia, Eastern Brazil, and Mexico. Few regions are identified as having no water-limitation meaning that while their growing season SIF-SM correlation is positive, it does not aid the fit in model estimation to have a non-zero slope. This likely means that the SIF-SM slope is positive, but near zero. Arid and semi-arid regions, with sparsely vegetated areas, show expected water limitation patterns. Among the pixels showing a water limitation on photosynthesis, 72.5% were characterized by a two-regime behavior, suggesting widespread nonlinearity of the soil moisture controls on vegetation. Therefore, at a given water-limited location, a unit loss of soil moisture typically confers more plant water stress when soil moisture is drier on average than when it is wetter.

Slope values are the highest (up to 10 [mW m$^{-2}$ nm$^{-1}$ sr$^{-1}$] per volumetric water content [m$^3$ m$^{-3}$]) in the Sahel region, Miombo woodlands south of the Congo Basin (Angola, Zambia, Mozambique), India, the Mekong Basin, and Eastern Brazil. These regions correspond well to the tropical climate, sub-climate savannah, of the Köppen-Geiger climate classification (Beck et al., 2018). In these regions, in the water-limited regime, a small incremental increase of soil moisture corresponds to a large increase in vegetation productivity and therefore the expected 8-day mean fluorescence emission. The high slope values of the regression between SIF and SM in drylands is mainly due to the clear relationship between photosynthetic efficiency, and therefore also $\phi_F$, and water availability. As a feedback of the increase in photosynthetic activity, the plant green biomass increases, leading to an increase in fPAR$_{Chl}$. The latter effects are especially determined by the water supply over drylands (Moreno-de las Heras et al., 2015). SIF observations allows monitoring of the combined biomass and photosynthetic efficiency effect. Values of the soil

moisture threshold are between 0 and 0.45 $m^3$ $m^{-3}$ with a median around 0.1-0.2 $m^3$ $m^{-3}$ (Fig. 6c and Fig. S3a for the

340  spatial distribution). When the soil moisture state is above this threshold, SIF has minimal to no water limitation. It is

worth noting that the threshold value might be harder to detect for regions with a low slope value. Furthermore, the

SM thresholds are correlated (across space) with soil texture (correlation coefficient $\rho = 0.37$, p-value $P < 0.01$ with

clay fraction; $\rho = -0.40$, $P < 0.01$ with sand fraction). Such correlations were similarly observed by Denissen et al.

(2020) over Europe. SM thresholds were also broadly assessed based on vegetation types using International

345  Geosphere-Biosphere Program (IGBP) land cover classification information. There is a tendency for the forested and

tree covered IGBP classifications to have higher soil moisture thresholds (Fig. S4a), though it is still unclear whether

the drivers of the soil moisture threshold are dominated by vegetation characteristics or characteristics of wetter mean

climate conditions.

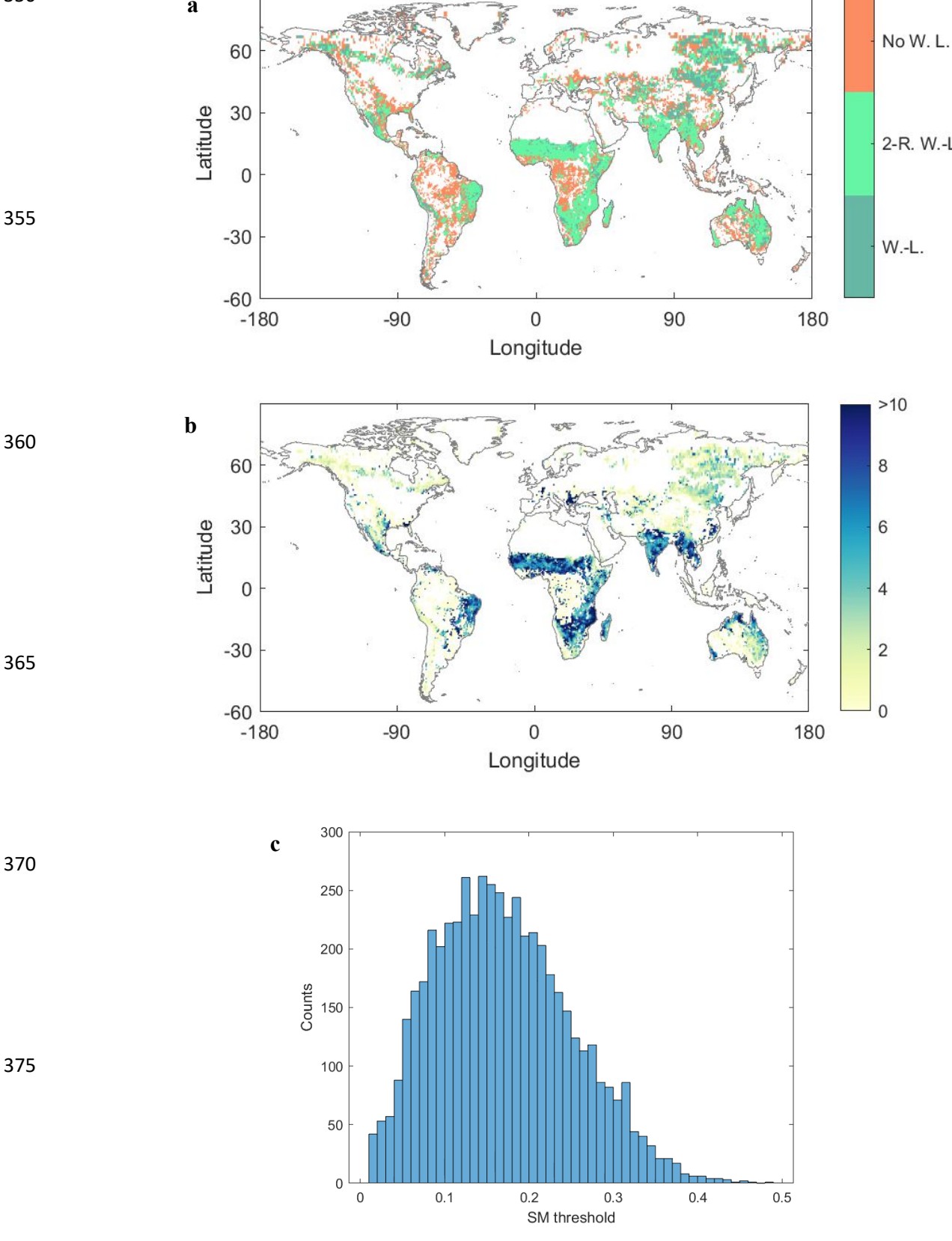

**Figure 6: Estimated SIF-soil moisture relationship features. (a) Model type (W.-L.: Water-Limited; 2-R. W.-L.: Two-Regime Water-Limited; No W. L.: No Water Limitation), (b) model slope [mW m$^{-2}$ nm$^{-1}$ sr$^{-1}$] in the water-limited regime, and (c) model threshold [m$^3$ m$^{-3}$] for the SIF-SM relationship. White shading denotes areas where the SIF-SM correlation is not positive—i.e., where SIF is not increasing with SM—or areas where no collocated SIF and SM records are available.**

## 4.2 Light-limited regimes

The spatial distribution of the selected model types, the corresponding model slopes, and the frequency distribution of model threshold for the SIF-PAR relationship are shown in Fig. 7. In contrast to the SIF-SM relationship, a light-limitation regime is observed in many parts of the northern hemisphere, mainly in Southern Canada, the Western U.S., the U.S. East coast, Western and Central Russia, the Balkans, and the Baltic region. Several regions are identified as having a break point between two regimes of light (non-) limitation, such as Western Europe (France, Spain, Italy, Great-Britain), Northern Russia, the U.S. Corn Belt, South-Eastern South America and South-Eastern Africa. Among the pixels showing a light limitation on photosynthesis, only 40.5% were characterized by a two-regime behavior. These regions of two-regime light limitation and threshold behavior are novel given that two-regime light influence on photosynthesis has not been observed or considered at large-scales previously.

Slope values are highest (up to 0.015 [mW m$^{-2}$ nm$^{-1}$ sr$^{-1}$] per [W m$^{-2}$]) in the midlatitudes, specifically in the Great Lakes regions of North America, most of Europe, Southern Russia, Northern Argentina, and Southern Brazil. These regions correspond well to the cold and temperate climates, sub-climate without dry season (hot or warm summer), of the Köppen-Geiger climate classification (Beck et al., 2018). A large proportion of these regions is used for annual crops. Their green biomass, and therefore the fPAR$_{Chl}$, is strongly affected by the (cumulative) PAR of the growing season. This explains a large part of the SIF-PAR relationship over these regions. In these regions, a small increment of light will substantially increase the vegetation productivity and therefore the expected fluorescence emission. This suggests that the Calvin cycle of these plants are adapted to strongly respond to light availability compared to other regions. By contrast, in the high latitudes of the northern hemisphere, slope values are the lowest, probably due to lower temperatures and limited seasonal changes in biomass of boreal ecosystems. Values of the estimated PAR threshold are between 0 and 140 [W m$^{-2}$] with a maximum occurrence around 100-110 [W m$^{-2}$] (Fig. 7c and Fig. S3b for the spatial distribution). When light availability is above this threshold, SIF has minimal to no light limitation. Such threshold behavior is theoretically expected based on the nonlinear relationship between incoming shortwave radiation and plant carbon fixation (Jones, 2014). Specifically, nonlinear relationships between SIF and PAR are expected

because many plant cellular processes are strongly light-limited in low light environments, but become maximized in brighter environments (Jones, 2014). For example, the light response curve fundamentally defines the rate of leaf-level carbon fixation that is limited under low light environments, but becomes Calvin cycle limited (i.e., carboxylation limiting) in brighter environments where more light on marginally produces more carbon uptake (Hermann et al., 2020). Laboratory experiments have shown a decrease in leaf-level fluorescence yield in high light environments (Wang et al., 2018). Similarly, De Cannière et al. (2022) observed a near-linear behavior of canopy-level SIF emission under low light conditions, while saturating at higher light values. Our threshold and slope estimates here are some of the first large scale observations of these fundamental light-limiting photosynthesis processes.

The PAR thresholds are additionally correlated with soil texture ($\rho = 0.25$, $P < 0.01$ with clay fraction). Significant differences in PAR threshold values are observed according to the IGBP land cover classes (see Fig. S4b).We note that PAR may relate strongly with surface temperature at seasonal scales in the northern hemisphere and thus relationships here may include the influence of surface temperature (Buermann et al., 2018; Zhang et a., 2020a). For example, the high SIF slope in the northern hemisphere midlatitudes may be inflated because we do not partition temperature limitation, which requires future investigation.

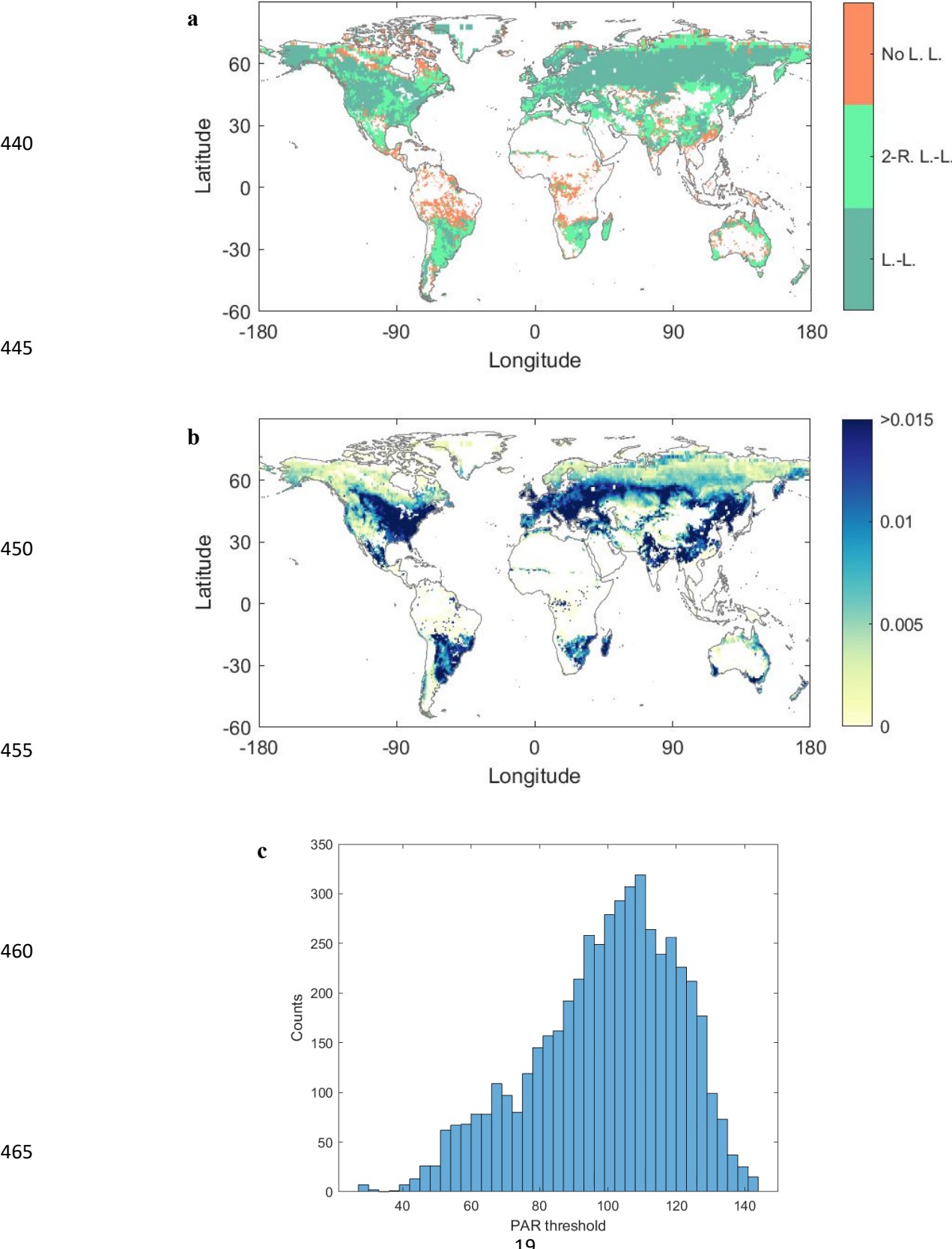

**Figure 7: Estimated SIF-photosynthetically active radiation relationship features. (a) Model type (L.-L.: Light-Limited; 2-R. L.-L.: Two-Regime Light-Limited; No W. L.: No Light Limitation), (b) model slope [$10^{-3}$ $nm^{-1}$ $sr^{-1}$] in the light-limited regime, and (c) model threshold [W $m^{-2}$] for the SIF-PAR relationship. White shading denotes areas where the SIF-PAR correlation is not positive—i.e., where SIF is not increasing with PAR—or areas where no collocated SIF and PAR records are available.**

### 4.3 Relationships of SIF limitation characteristics with mean moisture availability

SIF sensitivity to soil moisture shows a relationship with mean annual precipitation (Fig. 8a; $\rho = 0.32$, $P < 0.01$ with slopes corresponding to sloped part of the two-regime model and the one-regime linear model). Sensitivities peak at approximately 1,000 mm $yr^{-1}$. Locations with peak slopes occur in the wetter environments such as in India, Southeastern Asia, Angola, and Mozambique. These larger slopes are likely related to the degree to which vegetation responds to mean moisture and individual storms, given the weekly timescales of this analysis (Feldman et al., 2018). It also indicates that these wetter regions may have a stronger plant water stress response when the land surface becomes drier below the soil moisture threshold.

SIF sensitivity to PAR shows an even stronger relationship with annual precipitation (Fig. 8b; $\rho = 0.44$, $P < 0.01$), especially for regions below 1,000 mm $yr^{-1}$ (Fig. 8b; $\rho = 0.46$, $P < 0.01$). The increasing sensitivities may similarly be an adaptation of the vegetation to utilize light availability, given that moisture is typically less limited in these regions. Furthermore, both SM and PAR thresholds are correlated (across space) with mean annual precipitation (Fig. 8c and d; $\rho = 0.31$ and 0.29, respectively, $P < 0.01$; $\rho = 0.14$ and 0.37, respectively, $P < 0.01$, when considering only regions below 1,000 mm $yr^{-1}$).

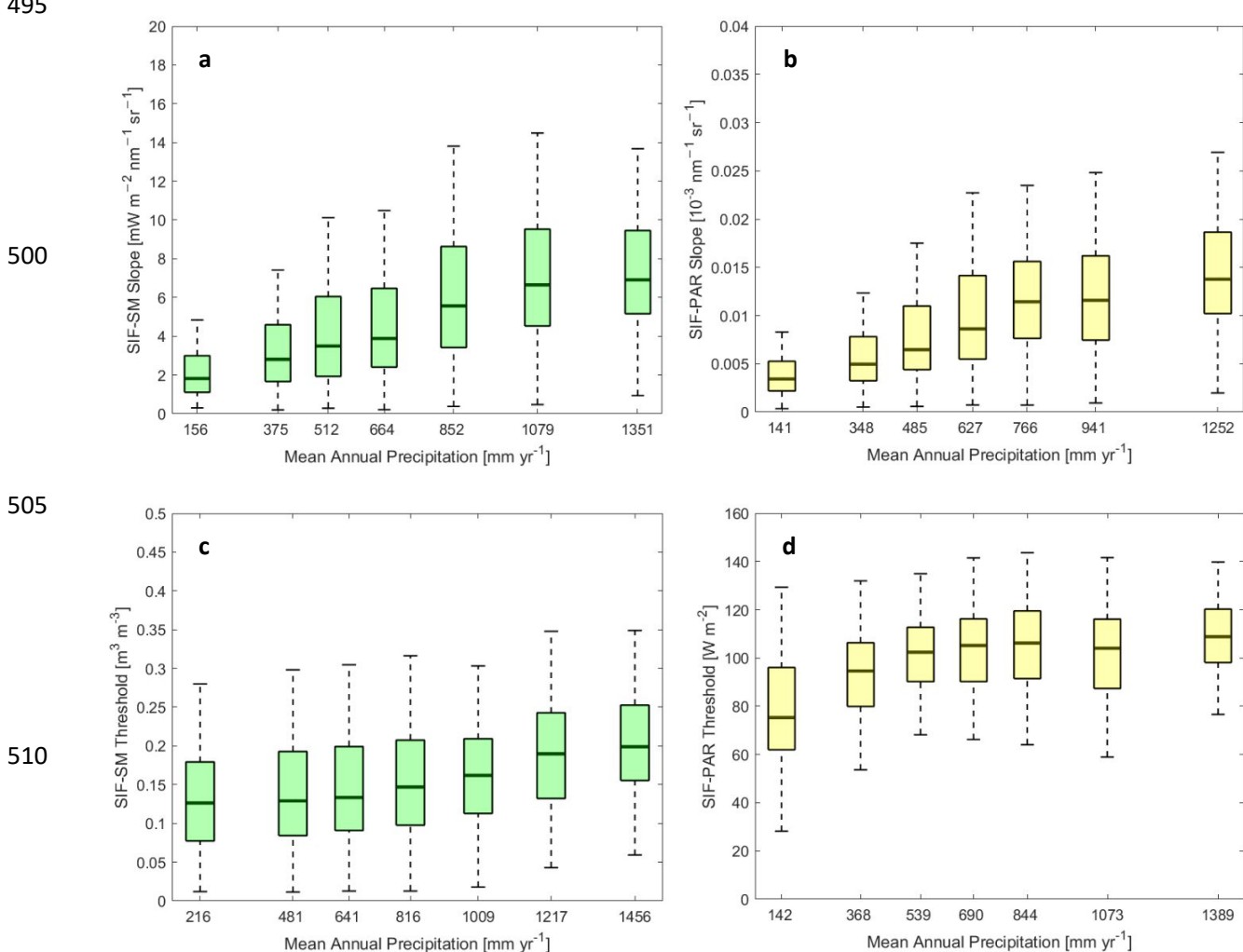

**Figure 8:** **(a) SIF-SM slopes [mW m$^{-2}$ nm$^{-1}$ sr$^{-1}$], (b) SIF-PAR slopes [10$^{-3}$ nm$^{-1}$ sr$^{-1}$], (c) SIF-SM thresholds [m$^3$ m$^{-3}$] and (d) SIF-PAR thresholds [W m$^{-2}$] binned as a function of mean annual precipitation [mm yr$^{-1}$] obtained from the GPM satellite constellation. Slope values correspond to sloped part of the two-regime model and the linear model (see Fig. 4). Each boxplot bin includes the same number of data points (915, 1548, 664 and 629 data points for boxplots (a), (b), (c), and (d), respectively). Box edges are the 25th and 75th percentiles of the distribution bounding the median (bold line), and whiskers extend to extrema (maximum and minimum). The median SIF-SM and SIF-PAR slopes and the median SIF-SM and SIF-PAR thresholds all increase with increasing mean annual precipitation.**

### 4.4 Robustness and Limitations

The main sources of uncertainty in this study include: (1) observation errors from the data streams (SIF, SM, PAR), (2) growing season definition errors, (3) model structural and parameter estimation errors, and (4) lack of consideration of all SIF-limiting factors. Here, we discuss each of these error sources and conduct several tests to evaluate the

robustness of results to these errors including repeating the analysis on a different satellite SIF dataset, on alternative growing season definitions and on deseasonalized variables.

To assess the effect of observation errors, we first repeat the analysis with GOME-2 SIF data, which resulted in comparable spatial patterns and thus robustness of the TROPOMI-based results (see Fig. S5). However, the main differences are reduced classification of the regimes with more parameters where the linear and two-regime models are selected less often. This is mainly because GOME-2 SIF results in fewer data pairs, which reduces the ability for the model selection to select more parameterized models. Another reason could also be related to the different timing of observations, with an overpass time near 13:30 local solar time for TROPOMI and near 9:30 for GOME-2. The higher water stress generally observed around noon compared to the morning could explain the lower detection of the water-limited regime (Qiu et al., 2020). TROPOMI was used as the main dataset given that its higher spatiotemporal coverage and lower retrieval noise were essential for the regime classification. We avoid evaluating alternative soil moisture and PAR datasets given that the study would result in a combinatorial analysis which we wish to avoid. Furthermore, it is more appropriate to evaluate alternative SIF measurements because SIF is a weak signal with relatively larger measurement errors given its retrieval from noisy atmospheric properties compared to lower error microwave remote sensing techniques for soil moisture, for example (Köhler et al., 2018; Jonard et al., 2020).

These results are based on a growing season with the start and end defined using the median NDVI (accounting for asymmetrical growing season with dynamic length). We found that spatial patterns of results were qualitatively the same when repeating the analysis considering a shorter growing season defined based on an NDVI threshold of 75% (see Figs. S7 and S8), which suggests a reduced impact of seasonality considerations on results. Therefore, the results are not a strong function of the growing season definition and are not greatly influenced by transitional time periods before and after the growing season that may be included in our definition. The fact that we condition on the growing season and do not assess the full year removes some influence of seasonality on the results, where too much influence of seasonality may amplify the determined connections between SIF with water and light. We acknowledge that this approach may miss secondary growing seasons in some regions.

Uncertainties from retrieval error can also be evaluated by calculating the coefficient of determination ($R^2$). However, $R^2$ is not appropriate for measuring the goodness of fit for nonlinear (here broken-linear) models. This is why the $R^2$ and the coefficient of variation (CV) of the model fit have been considered together as shown in Fig. S6. The $R^2$ values are higher in regions with high SIF sensitivity to SM or PAR (higher slopes in Figs. 6b and 7b). They tend to be lower in regions with more energy limitation (for SIF-SM), for example. This does not mean that the model fit is uncertain,

but that the SIF-SM slopes are approaching zero as physically expected. However, low $R^2$ values can be observed in other regions that do not conform to this claim. For example, boreal regions tend to have lower $R^2$ values, which may be related to instrument and/or retrieval noise. Some of these locations may have additionally low $R^2$ due to the simplified form of our models. However, we expect a large influence of SIF retrieval noise considering that retrieval error variance tends to be higher in SIF retrievals than for satellite-based vegetation reflectance metrics (Dechant et al., 2022). As such, $R^2$ values of 0.5 and 0.6 are relatively good fit given SIF retrieval noise that will limit higher $R^2$ values. An uncertainty analysis of the model selection and parameters using bootstrapping (with 1000 iterations) on 100 randomly selected pixels across the globe reveals that the mean bootstrapped standard deviation of the SIF-SM slope, SIF-SM threshold, SIF-PAR slope, and SIF-PAR threshold is 3.6 mW m$^{-2}$ nm$^{-1}$ sr$^{-1}$, 0.02 m$^3$ m$^{-3}$, 0.004 10$^{-3}$ nm$^{-1}$ sr$^{-1}$, and 5.5 W m$^{-2}$, respectively. The SIF-SM regime selection was the same more than 75% of the time for the 2-regime model, but the SIF-PAR regime selection tended to be more uncertain (same selection 60% of the time). This may be because the functional form of the SIF-PAR relationship is less-well defined in many locations than the SIF-SM functional form. An example of pixel is shown in Fig. S9. There is potential for structural errors and consequent parameter estimation errors in assuming a piecewise linear model fits a more complex nonlinear relationship that has curvature. As such, this creates uncertainties in soil moisture thresholds, for example. However, relatively low soil moisture threshold variances were found from the uncertainty analysis (mean bootstrapped standard deviation of 0.02 m$^3$ m$^{-3}$) showing robustness of the method and suggesting the relationship curvature may not bias thresholds greatly. Nevertheless, we emphasize the importance of the threshold detection, apart from its numerical value, which occurs despite the presence of curvature. The two-regime model is appropriate given known nonlinear relationships that drive environmental influence on plant function (Jarvis, 1976). As such, studies that do not acknowledge the different regimes and assume linear SIF relationships with the environment would bias estimations of SIF-SM or SIF-PAR relationships. Our results here suggest that with widespread detection of nonlinear SIF relationships with SM and PAR, the commonly used linear correlations and slopes between these variables, ignoring the nonlinearity, will create biases in evaluating water and light limitation. Finally, we repeated the results using the Akaike information criterion (AIC) rather than BIC to test the selection frequency of the more complex model forms (i.e., linear model and two-regime model). We fond that our use of BIC is conservative in selecting more complex models. AIC finds a much more frequent detection of more complex models at a greater risk of overfitting the data over that of BIC (not shown).

Not assessing all limiting variables including temperature and nutrient limitations is a drawback of the analysis. Our analysis indeed misses some limitations on plant function. For instance, no water- and no light-limitation can be seen

in the same region (such as tropical forests) where other bio-climatic factors (such as nutrient limitation) could

influence plant growth. Predominance of water versus light limiting regime might also shift over the growing season and between years, particularly in transitional climate regions (Seneviratne et al., 2010). However, we also note that the raw SIF-SM relationships, and not their deseasonalized relationships, include coupling from many factors beyond just water-limitation on SIF. We repeated the computation of linear slopes in the limiting regime using deseasonalized variables and found that the relationships are still positive, but are reduced (not shown). Because deseasonalized

variables will show a more direct, isolated connection between SIF and each of these limited variables, this analysis indicates that SM and PAR do have direct influences on SIF. However, the higher magnitude slopes in their raw interactions indicate that other factors and their interactions with water and light limitation are included in the SIF-SM and SIF-PAR relationships in addition to the influence of SM and PAR alone. As such, we argue in favor of determining the SIF-SM and SIF-PAR relationships with the raw (non-deseasonalized) variables to assess the state dependence and

coupling of multiple limiting factors on SIF. These overall relationships provide a test for model emergent behavior and coupled coevolution of multiple variables and their influence on SIF.

**4.5 Future Work**

Finally, we aimed to only detect and observe the emerging relationships in nature on how photosynthesis is limited by water and light. The procedure was purposefully naïve and required few assumptions, allowing the observations alone

to drive the results. This is a first step in the process of using detection procedures that include interactive effects of water and light with other variables. One can also obtain more mechanistic understanding of physiological behavior by estimating parameters, such as from Eq. (1), from the observed behavior in Figs. 6 and 7. However, these future approaches require more assumptions, which have the added challenge of detecting natural emerging behavior without biasing the results with restrictive assumptions.

The observation-only approach is chosen to best identify the naturally occurring and emerging plant function response to the environment. Such a study informs others that use model or reanalysis frameworks that have built-in assumptions that may confound the results. Nevertheless, the use of PAR from reanalysis here should be updated in the future with globally available shortwave radiation observations. Ultimately, we expect that MERRA-2 PAR is not largely influenced by built-in interactions with land surface behavior given that it is driven mostly by the atmospheric model

scheme and assimilation of soundings.

**5 Summary and Conclusion**

In this study, we map observational evidence for seasonal water-limitation and light-limitation in plant function at the ecosystem scale. We analyzed data from three different satellite sensors, namely sun-induced chlorophyll fluorescence from TROPOMI, surface soil moisture from SMAP, and normalized difference vegetation index from MODIS. In this purely observationally driven study, the combination of the three data streams allowed us to test a set of hypotheses on the types and extent of bio-climatic regions that should be classified as under seasonal water- or light limitation.

To detect where nonlinear controls of water and light on photosynthesis occur, three distinct models were tested representing three scenarios for each limitation (water and light). The first model is the linear model representing the water- or light-limited regime. The second model is the nonlinear two-regime model representing the situation where the rate limitation ceases above a certain threshold of soil moisture (SM) or photosynthetically active radiation (PAR). The conditions to select this model are conservative and thus we exhibit confidence in the detection of nonlinear controls when this model is selected. The third model is the zero-slope model for the no water or no light limitation regime.

The main results show that soil moisture limits on SIF are found primarily in drier environments while PAR limitations are found in intermediately wet regions. Nonlinear two-regime behavior is observed in 72.5% of the cases for water limitation on photosynthesis, while two-regime detection is much lower at 40.5% for light limitation on photosynthesis. Nevertheless, these nonlinear relationships are theoretically expected and widely observed across the globe for light limitation for the first time here. The widespread nonlinear control of water availability on SIF indicates that dry anomalies will differentially influence plant function: plants are buffered from reductions in water availability when soil moisture is higher, but will strongly respond to unit reduction in water availability under drought conditions. SIF sensitivity to PAR strongly increases along moisture gradients, reflecting mesic vegetation's adaptation to making rapid usage of incoming light availability on the weekly timescales investigated here. The transition point detected between the two regimes is connected to soil type and mean annual precipitation for both the SIF-SM and SIF-PAR relationships. These thresholds have therefore an explicit relation to properties of the landscape, although they may also be related to finer details of the vegetation and soil interactions not resolved by the spatial scales here. Future work can account for interactions between more variables and more explicit characterization of the nonlinear relationships in each pixel. Successful, systematic detection of nonlinear controls of individual environmental variables on photosynthesis with the statistical relationships is a first step here.

While our analysis is not exhaustive in not directly evaluating all possible factors (e.g., vapor pressure deficit (VPD), air temperature, nutrients) and their interactions, it highlights that vegetation function exhibits widespread, nonlinear dependencies on bio-climatic factors that are highly spatially variable. Given that we show vegetation existing in limited and non-limiting states depending on the water or light conditions, linear correlations of photosynthesis with specific resources provide limited views of landscape-scale photosynthesis. At the same time, many land-surface variables are tightly coupled and thus SM and PAR contain significant information about current meteorological conditions. Therefore, the information captured in bivariate SIF-SM and SIF-PAR relationships represents the real-world coevolution of photosynthesis with these limiting variables as they typically co-evolve with strongly-covarying temperature, VPD, etc.

As such, our study is unique in evaluating (1) the state-dependent, coupled controls on SIF, (2) in detecting the nonlinear relationships between plant function and water and light, major controls on global photosynthesis, and (3) in being an observational framework instead of using model-derived parameters. Our spatial maps therefore can serve as a benchmark to directly validate the model-emergent controls on terrestrial gross primary production from Earth system models.

**Special issue statement.** This article is part of the special issue "Microwave remote sensing for improved understanding of vegetation–water interactions (BG/HESS inter-journal SI)". It is not associated with a conference.

**Acknowledgments.** The authors acknowledge MIT for supporting this research with the MIT-Belgium seed fund "Early Detection of Plant Water Stress Using Remote Sensing".

**Data availability**. SMAP L2 soil moisture data are available from the National Snow and Ice Data Center (NSIDC, https://nsidc.org/data/SPL2SMP_E/versions/3). TROPOMI SIF data are available from Caltech Research Data Repository (CaltechDATA, https://data.caltech.edu/records/8hm1f-w5492). MERRA-2 PAR data are available from the Goddard Earth Sciences Data and Information Services Center (GES DISC, https://disc.gsfc.nasa.gov/datasets/M2T1NXLFO_5.12.4/summary). MODIS NDVI data are available from the Land Processes Distributed Active Archive Center (LP DAAC, https://lpdaac.usgs.gov/products/mod13c1v006/). Generated maps are available upon request.

**Author contributions**. FJ, AFF, DJSG, and DE conceived the study. DE led the project. FJ conducted the analysis and wrote the manuscript. AFF, DJSG, and DE contributed interpretations and numerous revisions to all versions of the manuscript, analysis, and figures.


**Competing interests.** The authors declare that they have no conflict of interest.

**Review statement**. This paper was edited by Julia K. Green and reviewed by René Orth and two anonymous referees.

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
