# Peer review of "Observed Water- and Light-Limitation Across Global Ecosystems"

_Biogeosciences, 2022_

## Author Response (AR1)

Reviewer #1: This study investigates the emergent relationship of vegetation functioning with water and energy availability across the globe. This is done largely with satellite-derived datasets, where vegetation functioning is characterized with Sun-induced fluorescence data and energy availability is represented by the photosyntetically active radiation. The authors test different linear models including breakpoints to show that the vegetation response to climate is non-linear in many areas for both energy and water limitation, as expected from physical principles.

Recommendation: I think the paper requires major revisions.

The topic of this study is interesting and timely. The response of vegetation to climate drivers is well known at small spatial scales from laboratory and field experiments, but it remains more unclear at larger spatial scales. At the same time, these vegetation-climate interactions are particularly relevant as they affect surface climate and need to be taken into account, and accurately captured by Earth system models, e.g. for projections of future climate scenarios. In this context, satellite-based datasets present an excellent opportunity to study these processes, and involved non-linearities and thresholds at model-relevant spatial scales.

I think that the manuscript is easy to read and understand and is a great match for the readership of Biogeosciences. However, before it is ready for publication some concerns regarding the analysis approach and robustness should be resolved, as detailed below.

Authors: Thank you for your constructive comments on our study. We addressed your comments as described in the responses below.

General comments:

(1) I think the fact that the seasonal cycles are not removed from the SIF, PAR and soil moisture data is a major shortcoming in this analysis. This way, confounding impacts by for example temperature can introduce artifacts in the results. For example, the positive correlation between soil moisture and SIF across a band across Canada in Figure 2 does not make sense from a physical point of view and could be related to such effects. I realize that the authors mention this point in section 4.3, and think it would be important if they could actually demostrate the negligible impact of the seasonal cycles by showing that similar results can be obtained with a shorter growing season of 1-2 months. Another way could be a synthetic experiment with e.g. evapotranspiration, radiation and soil moisture data from reanalysis where a similar time period can be used and the analysis can be done without removing the seasonal cycle (as here), and with the removal (while a representative seasonal cycle can be computed from many available years).

Authors R1: Indeed, if we were to deseasonalize the data streams, it would present a more direct connection between SIF and water and light. However, we do require the raw magnitude of each of the variables to determine the soil moisture threshold, the state dependence of which becomes obscured when using anomalies. Therefore, thresholds are best determined from the raw magnitudes. Instead, to reduce effects of seasonality, we perform the analysis during the growing season. In

response to the reviewer, we additionally performed the analysis over a shorter time period (3 months versus 6 months) in order to assess the relative impact of the seasonal effect. Figure R1 shows the results for the 3-month window that compares well to the 6-month window used in the original manuscript.

[Figure]

**Fig. R1: Estimated SIF-SM (top) and SIF-PAR (bottom) model slope in the water- or light-limited regime considering a 6-month (left) or 3-month growing season (right).**

As evident in the figure comparison, the magnitude of the slopes and the spatial patterns are similar. With the shorter window, we lose sample size but the magnitudes and patterns are comparable. Another reviewer suggested dynamic window size rather than fixed (examples 3 or 6 months). For example, using the period when NDVI is a certain percentage below its peak. In the revised paper, we take that approach rather than a fixed period and choose the median NDVI to define the start and end of the growing season.

We also reassessed the slope parameter estimates using deseasonalized anomalies of SIF, PAR, and soil moisture. In these analyses, we still need the mean magnitudes to find regimes – regimes are dependent on the magnitude of the state variables. Soil moisture and PAR thresholds were also determined from the analysis of the mean state (Figs. 6b and 7b). We then used those thresholds to estimate the slopes in the respective limiting regimes for PAR and SIF based on deseasonalized SM, PAR, and SIF (Fig. R2).

The fact that the slopes are of smaller magnitude indicates that water and light still directly influence SIF within our patterns. However, the additional slope magnitude likely captures other coupled interactions with soil moisture, for example. Ultimately, we choose to eliminate some seasonality via the growing season only approach, and present the raw magnitudes such that the relationships provide a validation for emergent model behavior of the influence of SM on SIF, for example, acknowledging that the relationships include interactions of SM with other variables.

See updates to our main text in Section 4.4 explaining these nuances in accounting for seasonality and evaluating the coupled influences on SIF (lines 571-581).

[Figure]

**Figure R2: Spatial distribution of the model slope for the (a) SIF-SM [m3 m-3] and (b) SIF-PAR [W m-2] relationships. For these maps, deseasonalized SM, PAR, and SIF were used, while SIF-SM and SIF-PAR thresholds and classifications displayed in Figs. 6 and 7 were considered.**

(2) There is no consideration of uncertainty for the performed analyses. In this context it would for example be informative to see some goodness-of-fit metric for the for the chosen model type (linear/two-regime/zero-slope), and possibly to introduce another category in case no reasonable fit was found for any of these types. Further, it would be interesting to which extent the model selection holds when for example synchronous bootstrapping would be performed for soil moisture and the other investigated variable. Moreover, in this context it would also be relevant to understand to which extent the results depend on the selected input datasets. I appreciate that the authors mention this aspect in section 4.3 in the case of alternative SIF datasets and think it would be important to demonstrate this in the supplementary material. Additionally, also the role of the soil moisture dataset should be tested by replacing the SMAP data with e.g. satellite-derived ESA-CCI soil moisture (https://www.esa-soilmoisture-cci.org/data) or upscaled in-situ soil moisture from the SoMo dataset (https://springernature.figshare.com/collections/Global_soil_moisture_from_in_situ_measurements _using_machine_learning_-_SoMo_ml/5142185).

Authors R2: We agree with the importance of considering the uncertainty in our results (and in results more broadly). Our goal here is to provide analyses of the spatiotemporal patterns of physical light and water limitation on SIF. We have now updated our manuscript with a new Section 4.4, which highlights several sources of uncertainty and directly shows how we handle each source of uncertainty in our analysis.

In summary, the new section includes a discussion of uncertainty of the statistical model. Our model (simple piecewise linear) is intentionally basic to allow readers to understand the model structural limitations. To quantify these, the coefficient of determination ($R^2$) and the coefficient of variation (CV) of the fit for each pixel are shown in the supplementary material (Fig. S5). In particular, $R^2$ is calculated for the linear model and the sloped part of the two-regime model, while the coefficient of variation is calculated for the constant part of the two-regime model and the zero-slope model where $R^2$ has no meaningful definition.

Observational errors are important as well, particularly when they are likely to be of similar or larger magnitude than the errors of using simple empirical methods. Additionally, following the reviewer's suggestion, the impact of the SIF dataset (TROPOMI vs GOME-2) is shown in the supplementary material (see Fig. S4 for results based on GOME2 SIF data) only to qualitatively provide an uncertainty

test on the overall spatial patterns of parameters rather than provide an analysis of differences of the datasets.

While it would be possible to perform the same analyses on different soil moisture data sets, we wish to avoid data sets that 1) significantly rely on modeled relationships between surface variables, 2) perform rescaling to match different sensors and microwave characteristics, or 3) will have major spatial representativeness uncertainties in comparison with aggregate SIF and PAR. This limits us to either the SMOS or SMAP datasets, which have a trade-off between a longer data record (SMOS) or low-level radio-frequency interference mitigation (SMAP). We selected SMAP in this case, particularly in the face of noisy data streams from satellite SIF observations. We don't feel that the addition of more SM data sets is likely to add physical insight, it will lead towards more of a combinatorial data set comparison study, which is not our intent. That said, SIF is both far noisier and a more complex proxy of ecosystem productivity status (as compared to satellite soil moisture), and we do see real benefit in terms of the robustness of response of physical relationships in assessing the impact of different SIF datasets rather than soil moisture datasets.

To evaluate the interactions between observation errors and our parameter estimation process, we could also perform a bootstrapping procedure to estimate the variance of the soil moisture threshold and slope on a randomly selected subset of pixels. This is of minimal utility to our main objective of giving first-order analyses of the spatiotemporal patterns of water and light limitation (rather than creation of a parameter data set) and we suggest that this is beyond the scope of the article. Finally, we find that our approach of model selection with BIC is conservative in selecting more complex models – repeating the approach with AIC presents more common detection of nonlinear relationships at the risk of overfitting the data (see our response to comment 3 below). As such, more confidence is portrayed in our approach when a more complex model like the "linear" or "two-regime" model is selected (see Fig. 4).

(3) I like that the authors recognize and determine non-linearities in the soil moisture-climate relationships. I think they could move a bit further in this direction by assessing the degree of non-linearity, for example as the difference in BIC scores between linear and non-linear models in each grid cell, or using the bootstrapping approach mentioned above. Further, it would be interesting to evaluate the spatial distribution of non-linearities as displayed in Figures 6a and 7a against climate and land surface characteristics, as done for the thresholds.

Authors R3: While we appreciate the path the reviewer suggests here, we also need to frame the purpose of such analyses in this study. We are using non-linearity as a model structural representation of limitation versus non-limitation regimes for SM and PAR. This is based on well-understood theory that if one variable is abundant, its marginal changes likely have no impact as a predictor (see for example citations below for thresholding relationships of some land surface variables). Our two-regime model then treats all limitation behavior as linear, which is oversimplified, but valuable as a first-order demonstration of the existence of limitation regimes, and to quantify how common each regime is for a location. The degree of nonlinearity will be a function of some landscape characteristics, but also of the local exogeneous forcings (e.g., the patterns of rainfall that might lead one location to switch between water and energy limited regimes). Disentangling these is not straightforward, and we wish to be careful to not oversimplify this for readers.

Furthermore, statistical tests for how nonlinear a relationship is rely on an assumption of the functional form of the nonlinearity, and therefore any metric for the "degree of nonlinearity" that would result from such a test is a large function of this assumed relationship. Our functional form (linear in the limitation regime, constant in the non-limitation regime) is the simplest possible for regime definition,

but not optimal for characterizing the full shape of a non-linear limitation relationship. To do so would require, 1) more theoretical understanding of what these shapes should be, and 2) a full sampling of the exogenous forcings space, which is not possible in "natural experiments" such as this.

Additionally, we stress that our current test is already effectively a test for whether the relationship is nonlinear. The current model (see Fig. 4) tests for whether or not linearity suffices to describe the relationships between SM and SIF and PAR and SIF. If a BIC score for the two-regime linear model is lowest, this means that linear behavior does not suffice and nonlinear behavior is detected. Differences in BIC scores are often used heuristically to compare goodness of fit between models, but do not have interpretable meaning in comparing one pixel to another (i.e., across data sets). Since inter-dataset comparison of information criteria is not possible, we propose to instead compute the AIC parameter to show whether another information criterion also detect nonlinearity (the existence of dual regimes) in the same pixels that BIC does.

Figure R3 shows the locations where AIC selects the two-regime model while BIC selects a one-regime model for both relationships. No location was found where BIC selects the two-regime model while AIC selects a one-regime model (not shown). As expected, BIC is more conservative than AIC, which will tend to penalize and reject higher-order (nonlinear) models to a greater degree. Therefore, AIC selects dual regimes (nonlinearity) more readily.

[Figure]

**Figure R3. Locations where AIC selects the two-regime model while BIC selects a one-regime model (black pixels) for the SIF-SM relationship (top) and the SIF-PAR relationship (bottom).**

References:

R. Akbar, D. J. Short Gianotti, K. A. McColl, E. Haghighi, G. D. Salvucci, and D. Entekhabi. Hydrological storage length scales represented by remote sensing estimates of soil moisture and precipitation. Water Resources Research, 2018.

E. Haghighi, D. J. Short Gianotti, R. Akbar, G. D. Salvucci, and D. Entekhabi. Soil and atmospheric controls on the land surface energy balance: A generalized framework for distinguishing moisture-limited and energy-limited evaporation regimes. Water Resources Research, 2018.

D. J. Short Gianotti, A. J. Rigden, G. D. Salvucci, and D. Entekhabi. Satellite and station observations demonstrate water availability's effect on continental-scale evaporative and photosynthetic land surface dynamics. Water Resources Research, 2019.

A. F. Feldman, A. Chulakadabba, D. J. Short Gianotti, and D. Entekhabi. Landscape-scale plant water content and carbon flux behavior following moisture pulses: from dryland to mesic environments. Water Resources Research, 2021.

A. F. Feldman, D. J. Short Gianotti, I. F. Trigo, G. D. Salvucci, and D. Entekhabi. Satellite-based assessment of land surface energy partitioning–soil moisture relationships and effects of confounding variables. Water Resources Research, 2019.

(4) I appreciate the analysis of the spatial patterns of the thresholds displayed in Figure S3 against climate and land surface characteristics. I think this analysis should additionally cover the role of the vegetation type (averaged across each grid cell), as this also affects the SIF-climate relationships. I realize this is mentioned by the authors in the conlusions section, and encourage them to include this into the analyses.

Authors R4: We added a figure assessing the spatial patterns of the model thresholds (1) as a function of vegetation types using IGBP land cover classification (Fig. S6), and (2) as a function of mean annual precipitation (Fig. 8). We discuss it more specifically in Sections 4.1 and 4.3 (lines 344-348 and lines 481-483).

(5) I did not quite understand why LAI was only obtained from a relatively short 4.5 year period only. The MODIS record extents further back in time, and a longer data record would allow to infer a more robust seasonal cycle.

Authors R5: In the revised manuscript, we revise to now use MODIS NDVI data from 2003 to 2020 to get the seasonal vegetation cycle to generate more confidence in the vegetation seasonality. See Sections 2.1.3 and 3.1 for the updated growing season determination with the longer record. Please note that this had minimal changes to our results.

Specific comments:

lines 38/39: this is not only true for increasing temperatures

Authors R6: We changed the text accordingly in lines 38-39.

line 41: what is meant with "rate-limiting"?

Authors R7: This has been removed from the text.

lines 42 & 62-68: the work from Li et al. (2021) is similar and relevant in this context and could be mentioned here

Authors R8: We added the reference Li et al. (2021) in the revised manuscript.

line 116: "day" should be singular, and a space should be removed in "Sentinel-5"

Authors R9:  This has been changed accordingly.

lines 142/143: what is the source of the soil texture data?

Authors R10: Texture data were obtained from the SoilGrids250m database. This has been added to the manuscript.

lines 169-171: Why not selecting the growing seasons as the 6 months with the highest LAI, independent on whether or not they would be consecutive, in order to better capture the highest LAI months in regions with more than one peak in the seasonal cycle?

Authors R11: There are many approaches described in the literature to define the growing season, but we believe that the approach we have followed is sufficient to characterize the active growing season encompassing the primary water and energy interactions with the carbon cycle. We additionally find that several alternative growing season definitions do not change our results. In the revised manuscript, we now take an alternative approach that uses determination of the peak LAI as well as accounting for asymmetrical growing season beginning and end times (rather than a fixed consecutive length). Specifically, rather than a fixed period and we used the median NDVI to define the start and end of the growing season. We acknowledge that this approach may miss cases of multiple growing seasons. See our updated text in Section 3.1 and uncertainty discussion of the growing season approach in Section 4.4 in lines 540 to 547.

line 177: replace "." with "x"

Authors R12: We replaced the multiplication sign with "x".

lines 291-296:  the work from Denissen et al. (2020) is similar and relevant in this context and could be mentioned here

Authors R13: We added the reference Denissen et al. (2020) in the revised manuscript.

lines 332-334: temperature limitation of vegetation functioning might play a role here, as mentioned also a few lines below

Authors R14: This has been clarified in the text (lines 401-402).

Figure S1:

Why is LAI in the mean seasonal cycles in b) and d) always the same across some consecutive days?

Authors R15: This is because LAI data were only available every 8 days and we realize how we plotted this originally may have been a misrepresentation of the dynamics. Figure S1 has been adapted as we use NDVI data from a longer time series of MODIS observations in the revised manuscript, which now corrects this issue.

Reviewer #2: Jonard and colleagues identify regions globally whos productivity during the growing season is (temporarily) limited by water or light availability. Based on S5-p Tropomi SIF, SMAP soil moisture, MERRA shortwave incoming radiation and MODIS LAI they identify whether the SIF in a pixel is constantly limited, constantly unlimited by water/ light, or whether a break point exists, where SIF changes from limited to unlimited behaviour. They find large regions for both the water and the light limited regimes which exhibit a breakpoint, claiming that the assumption of a linear relationship to SIF as in many studies does not hold. Particularly interesting is the identification of a breakpoint for the light limitation in many pixels. The locations of the breakpoints along gradients in soil moisture and light show a relationship with mean precipitation and soil type.

The authors have done a good job in outlining the scope of their study and the aspects that render it unique from related work in the literature. The methodological approach is overall sound and limitations openly discussed, except for one aspect related to the definition of the growing season that I will outline in more detail below. In my opinion, there are several aspects that need to be included (at least in the discussion, if not in also in the analysis), and they will render it more convincing (see below). The topic is relevant and fits Biogeosciences, I suggest publication after addressing the following points:

Authors: Thank you for your constructive comments on our study. We addressed your comments as discussed in the responses below.

1) Given that it is a purely observational study, the limitations of the remotely sensed data streams at the basis of the analysis need to be taken into account and discussed, which so far has not happened. I suggest to include in the manuscript information on whether there is quality control applied to the observations. Are there regions where the observational coverage is critically low during the growing season, such that uncertainty in the selected model becomes large? How is persistent cloud cover during the wet season handled, do I understand correctly that with the given definition of the growing season we rather evaluate the cloud-free dry down period in seasonally dry ecosystems? SIF penetrates through partial cloud cover, but persistent full cloud cover is critical as well. Please note, that LAI is not purely observational but also based on a model.

The different data streams cover different periods, why is that?

Authors R1: We added in the methods section information on quality control applied to the observations (e.g., filtering of pixels with large cloud cover). See Section 2.1.1. We also clarify that we screen pixels that have less than 20 data pairs such that an adequate model selection can be performed (lines 304-305). Fig. R1 shows a map of the number of data pairs in each pixel that are used in the analysis.

We agree that LAI is not purely observational and we now use NDVI, a more observational metric without the influence of model assumptions, in the revised manuscript. Please see our updates in Section 2.1.3 and Section 3.1 that now use NDVI.

*We clarified in the text that we used the same data period for the main analysis (SIF, SM and PAR data streams). However, we use longer datasets based that we need to determine average climatological information such as LAI (or now NDVI) climatology (2003-2020) and GPM annual mean precipitation (2010-2020). See the beginning of Section 2 where we clarify this point.*

[Figure]

**Fig. R1: Number of data pairs in each pixel that are used in the analysis for the SIF-SM (top) and SIF-PAR (bottom) relationships. Inset shows the histogram of the number of data pairs per pixel.**

2) The growing season is defined as the 6 months around the annual peak LAI, and the authors tested other definitions of the duration of the growing season (more details on which ones those are?) with qualitatively similar results. But I wonder to what extent regions with different length of the growing season can be compared to each other based on a period with fixed duration around the peak of LAI. Depending on the actual length of the growing season (I mean between the start of the green-up and the end of the greenness period), the 6 (or x) months around the peak will cover a different fraction of the growing season in different pixels (e.g. in higher latitudes or arid regions with only a short wet period, the 6 months might include a rather large fraction of the shoulder seasons or even the non-growing season, which will not be the case for temperate regions). As you discuss as well, the main driver of productivity can change over the course of the growing season, and also the year. So how can this affect your results? What about potential alternative definitions, such as to focus the

analysis on the peak of the growing season of each individual pixel, defined as e.g. half/30% or similar of the duration between start of green-up and end of green period around the peak LAI (neglecting potential asymmetry in the growing season)?

Authors R2: We do agree the 6-month definition is a bit arbitrary and can include variable amounts of the times outside of the growing season, confounding one-to-one comparison between pixels across the globe. Similarly to what the reviewer suggests, we now instead define the growing season by first finding the peak NDVI (95th percentile) to find the main growing season and then finding the green up and brown down times as when NDVI reaches its median before and after this peak. There are many potential methods of finding the growing season, each with their own shortcomings. We additionally test other definitions of shorter growing seasons of 3 and 4 months (Fig. R2). While these alternative definitions reduce the number of data pairs available for the per-pixel analysis, the spatial patterns of our results do not change.

[Figure]

**Fig. R2: Estimated SIF-SM (top) and SIF-PAR (bottom) model slope in the water- or light-limited regime considering a 4-month (left) or 3-month growing season (right) centered on the NDVI peak.**

3) It is well appreciated that the authors tested their approach with other SIF data sets as well and come to qualitatively similar results. Although the results are not shown (which is fine, but would be nice to see in the SI), to me it would be important to clarify in the manuscript how this was done. Did you use the same gridding, and also the same years, or did you exploit the longer records of OCO2 and GOME2 SIF and did the analysis for more years, potentially more robustly? Does the fact that the study period focussed on the particularly dry years in central Europe in 2018/19 affect your conclusions for this region?

Authors R3: We agree with the reviewer's comment. Results based on GOME-2 SIF data are shown in Fig. S4 for a 4-year period from April 2015 to March 2019 (last period for which GOME2 SIF data are available). See our description of the data in Section 2.1.1. We find that the spatial patterns are similar to those of TROPOMI, but with less available data in GOME2. This GOME-2 nevertheless analysis acts as a test of robustness which we discuss in new Section 4.4 in lines 529 to 538.

4) The revelation of the two-regime behaviour for light limitation is something particularly novel in the manuscript. Its discussion is very short, I strongly suggest extending the discussion of this point and the related thresholds.

Authors R4: We agree. We have added more discussion about the two-regime behaviour for light limitation to place it into context with theory. See section 4.2 in revised lines 405 to 415.

5) I like the summary of related literature in the introduction very much. However, one aspect goes missing in my opinion, and that is the aspect of the time scales. The studies cited focussed on different time scales, and the main drivers of productivity might change across time scales (Linscheid et al. 2020, Biogeosciences, 10.5194/bg-17-945-2020), and this aspect should be clarified in the introduction.

Authors R5: This has been clarified in the Introduction in lines 97 to 100.

6) The last part of the conclusions is missing. Parts of the conclusion read very similar to parts of the abstract.

Authors R6: We have revised the conclusion to give more context, for example, to the nonlinearity findings results and what they mean for drought response as well as why these results can be used to evaluate emergent behavior from global models.

---

## Author Response (AR2)

**Reviewer #1** (Second review of Jonard et al., bg-2022-25)

I appreciate the efforts made by the authors to address my and the other reviewers' comments which have improved the paper. At the same time I believe that some of the revisions need to be refined and/or expanded before the paper can be published:

-- regarding main comment (1) from my previous review

I appreciate the additional analyses implemented by the authors to test the effect of the seasonal cycles on the results. Also, I agree with the new definition of the growing season as the time of year between peak and median NDVI. Building upon this I would suggest to perform the SIF-SM and SIF-PAR analyses also using a shorter growing season (as they do in the rebuttal) but to define this in a consistent way with the main analysis and to only change the NDVI threshold from median to e.g. 75%. This will more effectively remove seasonality than simply reducing the growing season length to 3 or 4 months. Moreover, a more quantitative assessment of the agreement of the resulting spatial patterns (through e.g. correlations between maps) in terms of the slopes and regimes is needed rather than concluding that they are similar (this is also valid for the results obtained with GOME-2 SIF as provided in Figure S4). Finally, these results also need to be added to the supplementary material to inform all readers.

Authors R1: We performed the analysis using a shorter growing season defined based on an NDVI threshold of 75%, as suggested by the reviewer. The results for the model type, model slope, and model threshold for both SIF-SM and SIF-PAR relationships using this new definition of the growing season are shown in Fig. R1 and R2 and are also added in the supplementary material (see Figs. S7 and 8).

As suggested by the reviewer, we also compared the regime classifications for both growing season definitions (NDVI threshold of 50 and 75%). As you can see in Fig. R3a and b, the majority of pixels (64% and 56%, respectively) have the same regime classification for both methods (green pixels). Looking at the SIF-SM relationship, as we move to shorter growing season, we increasingly get more detection of the zero-slope regime and less detection of the linear sloped regime. By constraining more and more to a part of the growing season, we detect the energy-limited regime more and more. This reduction of match of regimes between the growing season definitions is therefore more explained by constraint to energy-limited regime (a physical aspect we expect) rather than an artifactual one due to seasonality considerations.

[Figure]

Fig R1. Estimated SIF-SM relationship features based on a shorter growing season defined using an NDVI threshold of 75%. (a) Model type, (b) model slope [mW m$^{-2}$ nm$^{-1}$ sr$^{-1}$] in the water-limited regime, and (c) model threshold [m$^3$ m$^{-3}$].

[Figure]

Fig R2. Estimated SIF-PAR relationship features based on a shorter growing season defined using an NDVI threshold of 75%. (a) Model type, (b) model slope [$10^{-3}$ nm$^{-1}$ sr$^{-1}$] in the light-limited regime and (c) model threshold [W m$^{-2}$] for the SIF-PAR relationship.

[Figure]

Fig R3. Comparison of model selection for the SIF-SM (top figure) and SIF-PAR (bottom figure) relationships considering two different growing season definitions, i.e., based on the 50th and 75th percentile of NDVI. Locations where the selected model is the same for both definitions are shown in green, while locations where the selected model is different are shown in orange.

-- regarding main comment (2) from my previous review

Also for this point I appreciate that the authors have taken mine and others' comments serious and have implemented several additional analysis which provide more insights into the limitations of their study.

However, I do not agree with the statement that R2 are "well above 0.5 in the regions where linear and two-regime models are found" in line 549. Furthermore, I would suggest to generally exclude grid cells from the analysis where linear or two-regime models are detected but the R2 is low, e.g. below 0.2.

Authors R2: We think that $R^2$ is not the best metric to screen pixels of the maps presented in Figs. 6 and 7 (see main manuscript). In particular, $R^2$ does not allow to assess the goodness of fit for a brokenlinear (non linear) model. Instead, this is why we emphasize that the R2 and coefficient of variation metrics need to be considered together as shown in Fig. S6. Furthermore, we want to show all the information and not limit to an arbitrary $R^2$ threshold. We changed the sentence by (lines 560-572):

"However, $R^2$ is not appropriate for measuring the goodness of fit for nonlinear (here broken-linear) models. This is why the $R^2$ and the coefficient of variation (CV) of the model fit have been considered together as shown in Fig. S6. The $R^2$ values are higher in regions with high SIF sensitivity to SM or PAR (higher slopes in Figs. 6b and 7b). They tend to be lower in regions with more energy limitation (for SIF-SM), for example. This does not mean that the model fit is uncertain, but that the SIF-SM slopes are approaching zero as physically expected. The coefficient of variations being low corroborates acceptable model fit in these cases, such as in the Congo and Amazon Basins. However, low $R^2$ values can be observed in other regions that don't conform to this claim. For example, boreal regions tend to have lower $R^2$ values, which may be related to instrument and/or retrieval noise. Some of these locations may have additionally low $R^2$ due to the simplified form of our models. However, we expect a large influence of SIF retrieval noise considering that retrieval error variance tends to be higher in SIF retrievals than for satellite-based vegetation metrics or modeled leaf area index (Dechant et al., 2022). As such, $R^2$ values of 0.5 and 0.6 are relatively good fit given SIF retrieval noise that will limit higher $R^2$ values."

Dechant, B., Ryu, Y., Badgley, G., Köhler, P., Rascher, U., Migliavacca, M., Zhang, Y., Tagliabue, G., Guan, K., Rossini, M., Goulas, Y., Zeng, Y., Frankenberg, C., and Berry, J. A.: NIRVP: a robust structural proxy for sun-induced chlorophyll fluorescence and photosynthesis across scales, Remote Sens. Environ., 268, 112763, 10.1016/j.rse.2021.112763, 2022.

Regarding my suggestion to perform bootstrapping to determine the uncertainty of the regime and threshold classifications I would like to clarify that this should be performed *in time* rather than *in space* as the authors described it in the rebuttal. Anyway, I do not fully agree with the arguments of the authors for not performing the bootstrapping, but if they would like to avoid this I would suggest to tone down the statements on the regime and threshold identification in the manuscript where e.g. now in the abstract and conclusions exact numbers of the occurrence of two-regime behavior are given while in some grid cells another regime/model would probably fit almost equally well (which the bootstrapping would allow to test).

Authors R3: We performed the bootstrapping analysis on 100 pixels randomly selected from the pixels classified as a 2-regime model in Fig. 6(a) of the main manuscript. We randomly generated the locations of the selected pixels, which results in a distribution across the globe.

We added the following text in the results section (lines 574-580): "An uncertainty analysis using bootstrapping (1000 iterations) on 100 randomly selected pixels across the globe revealed that the mean bootstrapped standard deviation of the SM-SIF slope, SM-SIF thresholds, PAR-SIF slopes, and PAR-SIF thresholds are 5.2 $mW\ m^{-2}\ nm^{-1}\ sr^{-1}$, 0.02 $m^3\ m^{-3}$, 0.004 $10^{-3}\ nm^{-1}\ sr^{-1}$, and 5.5 $W\ m^{-2}$, respectively. The SM-SIF regime selection was the same 76% of the time for the 2-regime model, but the PAR-SIF regime selection tended to be more uncertain (same selection 60% of the time). This may be because the functional form of the PAR-SIF relationship is less-well defined in many locations than the SM-SIF functional form. An example of pixel is shown in Fig. S9."

[Figure]

Fig R4. Uncertainty analysis of the model selection, model slope and model threshold using bootstrapping (1000 iterations) for two representative locations. Left column: pixel in the Central African Republic (latitude/longitude: 9.65°N/22.78°E) showing a two-regime water limitation. Right column: pixel in Argentina (latitude/longitude: 32.79°S/63.11°W) showing a two-regime light limitation.

Minor comments:

- line 43: It should be Li et al. 2021

Authors R4: This has been corrected.

- lines 303 and 563: BIC and AIC could be introduced a bit more by adding some information on the motivation of choosing them and their characteristics; for AIC not even the abbreviation is explained

Authors R5: We added the description of the AIC abbreviation in the revised version (line 592). We also added the following text to provide more information and motivation for choosing BIC (lines 309-314): "Specifically, BIC penalizes more complex models (i.e., two regime model here) that inherently increase the model fit to the data, but may not provide more predictability than less parametrized models. For example, the linear model has one additional parameter than the zero-slope model, which is the slope. In order for the linear model to be selected, the additional parameter of a non-zero slope must increase the model fit beyond the penalty of overfitting. Note that information criterion like BIC have been applied to similar model selection applications of water and energy limitation (Feldman et al., 2019; Schwingshackl et al., 2017)."

- line 596: sumary --> summary

Authors R6: This has been corrected.

- it would be helpful to have an overview table for all employed datasets to quickly access information

on their native space-time resolutions

Authors R7: We have added the following table:

| Variable | Source of data | Spatial resolution | Temporal resolution |
|---|---|---|---|
| Sun-induced chlorophyll fluorescence | Sentinel-5P satellite TROPOM instrument | 7 x 3.5 km² | daily |
| Soil moisture | SMAP satellite L-band radiometer instrument | 36 x 36 km² | 3 days |
| Normalized Difference Vegetation Index | Terra satellite MODIS instrument | 0.05° | 16 days |
| Photosynthetically active radiation | MERRA-2 global reanalysis | 0.5° x 0.625° | daily |
| Precipitation | GPM satellite constellation IMERG product | 0.1° x 0.1° | Half-hourly |

Table R1. List of datasets used in this study with their respective native spatial and temporal resolution. Note that the datasets were all linearly aggregated to a spatial resolution of 72 x 72 km² and 8-day periods.

- color bars in Figure S5b,d should be adapted to avoid the inclusion of negative values

Authors R8: This has been changed accordingly.

- Figure S6b (which is nice!) is not referred to in the text

Authors R9: A reference to the figure (called now Fig. S4b) has now been added (lines 425-426).

**Reviewer #2**

Jonard et al. presents an observational study to investigate the limitation of light and water on ecosystem photosynthesis across the globe. They consider three types of models to characterize the light and water limitation: water/light-limited, two-regime water/light-limited, and no water/light limitation. Since I received the revised version of manuscript, I found that the authors carefully considered all the comments by two reviewers and addressed most of their comments. They performed several additional analyses to support the robustness of their study, including deseasonalizing the data stream, considering different definitions of growing season, highlighting several sources of uncertainty, testing the difference of BIC and AIC method for model selection, replacing LAI by NDVI, assessing the model thresholds separately for different vegetation types etc. The current manuscript is largely improved compared to the original one. I only have some minor comments. Therefore, I suggest that it can be published in Biogeosciences after addressing the following points.

Authors: Thank you for your constructive comments on our study. We addressed them as described in the responses below.

Minor comments:

1) Figure R2: The spatial pattern of (a) is visually similar to that of Fig. 6 (b). May use a different colorbar for better comparison.

Authors R1: Now, a shorter growing season defined based on an NDVI threshold of 75% is used instead of a 3- or 4-month growing season, as suggested by Reviewer 1. We think that this would allow a better comparison with the Fig. 6 and Fig. 7 of the main manuscript.

2) Figure S4: The two-regime water-limited regions identified by GOME-2 were almost captured by TROPOMI, while TROPOMI identified much more widespread two-regime water-limited regions than GOME-2. As authors mentioned, the fewer data pairs, coarser spatial resolution, higher retrieval noise of GOME-2 SIF could partly account for it. How about other reasons that can also be discussed? I found most of these regions (two-regime by TROPOMI) were classified as no water limitation by GOME-2. Is this also related to the method that characterize different regimes? For example, it's not very clear how to define the 'zero-slope' model: should the slope strictly be zero? or how to distinguish 'zero-slope' model with a very small slope but linear model (water-limited). If I missed these texts, please point it out.

Authors R2: We have added more detailed text with regard to model selection and the specific example difference between selecting the linear model with a non-zero slope and the zero-slope model. See lines 310-314.

3) One other reason may be the difference in overpass time of TROPOMI and GOME-2. The overpass time of TROPOMI is around 1 pm local time while 10:30 am for GOME-2. The ecosystems usually have higher water stress at midday compared to the morning. Please see the highly asymmetrical diurnal cycle of GPP for some dryland sites (Fig. 3c in Qiu et al., 2020 and Fig. 8 in Li et al., 2021). Therefore, this could be another reason why GOME-2 detected less water-limited regions.

Qiu B, Ge J, Guo W, et al. Responses of Australian dryland vegetation to the 2019 heat wave at a subdaily scale. Geophysical Research Letters, 2020, 47(4): e2019GL086569.

Li X, Xiao J, Fisher J B, et al. ECOSTRESS estimates gross primary production with fine spatial resolution for different times of day from the International Space Station. Remote Sensing of Environment, 2021, 258: 112360.

Authors R3: Thank you for the comment and the interesting discussion on the impact of the different timing of observations between TROPOMI and GOME-2. We included the following sentence in the new version of the manuscript: "Another reason could also be related to the different timing of observations, with an overpass time near 13:30 local solar time for TROPOMI and near 9:30 for GOME-2. The higher water stress generally observed around noon compared to the morning could explain the lower detection of water-limited regime (Qiu et al., 2020)." See lines 541-544.

4) I understand the authors do not want to include combinatorial results from multiple SIF, SM, and PAR datasets. Considering that a) SM is one of your most important data; b) even using different SIF data could slightly to moderately change the results, I also agree with reviewer 1' suggestion and prefer to consider at least one more SM data. If do so, may not need to perform all the analyses by a new SM data or present combinatorial results, and I think only one key figure in supplementary material is enough (like Fig. 6b). But for this comment, I respect the authors' own decision.

Authors R4: We confirm that we don't think that the addition of more SM data sets is likely to add physical insight, it will lead towards more of a combinatorial data set comparison study, which is outside the scope of this study.

Lines 561-562: 'not a strong' and 'not greatly' still indicate that your results are (may moderately) affected by the different definition of growing season.

Authors R5: We agree, the different maps do not match perfectly but the general patterns are the same, which suggests a reduced impact of seasonality considerations on results. This has been added to the text (line 554).